biomedical engineering/biomechanics

knee adduction moment, knee medial loading, musculoskeletal modelling, neuromuscular electrical stimulation

**Author for correspondence:**
Dong Ming
e-mail: richardming@tju.edu.cn

# Extra excitation of biceps femoris during neuromuscular electrical stimulation reduces knee medial loading

Rui Xu[1,2], Dong Ming[1], Ziyun Ding[2]
and Anthony M. J. Bull[2]

[1]Department of Biomedical Engineering, College of Precision Instruments and Optoelectronics Engineering, Tianjin University, Tianjin 300072, People's Republic of China
[2]Department of Bioengineering, Imperial College, London SW7 2AZ, UK

RX, 0000-0003-2298-7582; DM, 0000-0002-8192-2538;
ZD, 0000-0002-1400-792X; AMJB, 0000-0002-4473-8264

Medial knee joint osteoarthritis (OA) is a debilitating and prevalent condition. Surgical treatment consists of redistributing the forces from the medial to the lateral compartment through osteotomy, or replacing the joint surfaces. As the mediolateral load distribution is related to the action of the musculature around the knee, the aim of this study was to devise a technique to redistribute these forces non-surgically through changes in muscle excitation. Eight healthy subjects participated in the experiment, and neuromuscular electrical stimulation was used to change the muscle forces around the knee. A musculoskeletal model was used to quantify the loading on the medial compartment of the knee, and a novel algorithm devised and implemented to simulate neuromuscular electrical stimulation. The forces and moments at the knee, ground reaction forces, walking velocity and step length were quantified before and after stimulation. Stimulation of the biceps femoris resulted in a significant decrease in the second peak of the medial knee joint loading by up to 0.17 body weight ($p = 0.016$). Kinematic parameters were not significantly affected. Neuromuscular electrical stimulation can decrease the peak loads on the medial compartment of the knee, and thus offers a promising therapy for medial knee joint OA.

# 1. Introduction

Osteoarthritis (OA) is a major degenerative disease, the prevalence of which is predicted to increase significantly due to a growing

ageing population [1]. The highest prevalence of OA is in the medial compartment of the knee [2], and excessive contact force is known to be a risk factor for the onset and progression of medial knee OA [3]. Therefore, reducing the contact force on the medial compartment of the knee is a tantalizing opportunity to treat medial knee OA.

It has been shown that gait modification by varying the external forces on the knee through valgus bracing and lateral wedges can decrease the knee adduction moment (KAM) and/or shift the centre of pressure of the ground reaction force (GRF) laterally [4–7]. These measures are surrogates of the medial knee contact forces (KMF) and the efficacy of these approaches is highly dependent on the individual subject [8,9].

As the forces acting upon the knee comprise external forces that are counterbalanced by the internal forces (reaction forces between the bony segments, muscle forces and ligament forces) [10], and internal forces are a redundant system with multiple different muscle contraction patterns able to provide this counterbalance, so it is conceivable that by changing the internal muscle forces, the mediolateral load distribution can be modified without changing the external forces. This distribution is highly related to the action of the hamstrings, quadriceps and gastrocnemius muscles [11]. For example, it is known that an elevated KAM is directly correlated with increased medial loading, and that quadriceps and gastrocnemius offer the most resistance to KAM [12]. Therefore, an alternative to orthotic gait modifications is to directly change the muscle forces during gait.

Muscles forces can be changed by altering the timing and strength of contraction through, for example, physiotherapy. An alternative approach that is used for quadriceps muscle strengthening for knee OA patients, when they present with chronic pain and joint stiffness [13,14], is neuromuscular electrical stimulation (NMES). NMES is a non-invasive technology that induces a voltage gradient along axons that can depolarize membranes that then induce action potential, which causes muscle contraction. NMES of the quadriceps has been shown to modify knee moments [15] and could be used to reduce KMF by selecting the appropriate muscles for stimulation. Muscles known to realign the distribution of the medial and lateral knee forces are the long head of biceps femoris (loBF), lateral gastrocnemius (latGAS) and vastus lateralis (VL) [11]. Therefore, in this study, it is hypothesized that the selective excitation of these muscles will redistribute the joint loading from the medial to the lateral side in healthy subjects, and thus, NMES may be used to ameliorate symptoms in sufferers of medial knee joint OA.

KMF can be measured directly through the use of implanted prostheses [16], partially inferred through the KAM [17–21], are indirectly related to knee flexion moment (KFM) [21,22], and can be reliably estimated through the use of musculoskeletal modelling [23–26]. Of these, the only method suitable for non-invasive direct quantification of KMF in those without arthroplasty is musculoskeletal modelling. Some musculoskeletal models are driven by electromyography (EMG) measurements; however, NMES stimulus contaminates EMG [27] meaning that this approach cannot be used. Other methods are based on static optimization [28–31] that use cost functions that minimize muscle fatigue [32], defined as the sum of cubed muscle stresses. However, when NMES is used to externally drive muscle recruitment, the cost function is not representative. Therefore, in this study, a modified static optimization cost function is proposed to estimate KMF in NMES-assisted gait.

The aim of this study was to test the hypothesis that extra excitation of the lateral leg muscles (i.e. loBF, latGAS and VL) can reduce KMF during level walking, where KMF is quantified through the use of the newly proposed cost function in a validated musculoskeletal model [23]. This is then compared with the commonly used surrogate of KMF, the KAM, and finally, the input kinematic and kinetic data are analysed to explain the results.

# 2. Methods

## 2.1. Experimental data

Eight healthy subjects (age, $27.3 \pm 2.4$ years; mass, $62.7 \pm 14.4$ kg; height, $1.69 \pm 0.11$ m; three males and five females) participated in this study which consisted of performing four gait tasks, three trials each of: normal walking, and three walking tasks in which separate muscles were stimulated individually (loBF, latGAS and VL).

The skin overlying the muscles was prepared with 70% isopropyl alcohol skin wipes, and the electrode pads (PALS® Platinum, Axelgaard Manufacturing Co., Ltd, Fallbrook, CA, USA) were placed at the two motor points of the relevant muscles (figure 1a). An asymmetrical biphasic waveform was used with

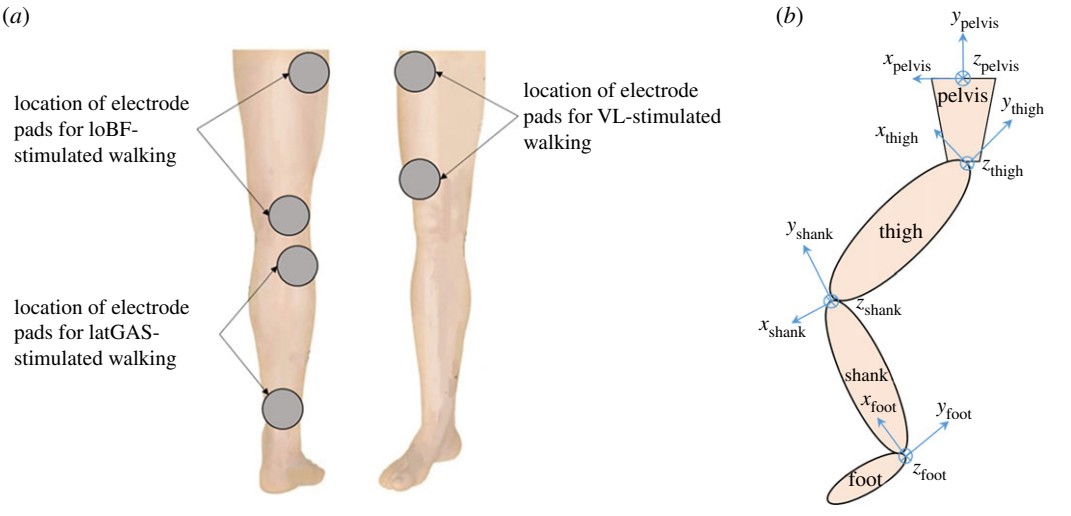

**Figure 1.** (*a*) Electrode pad positions for stimulated walking. (*b*) Local coordinate systems.

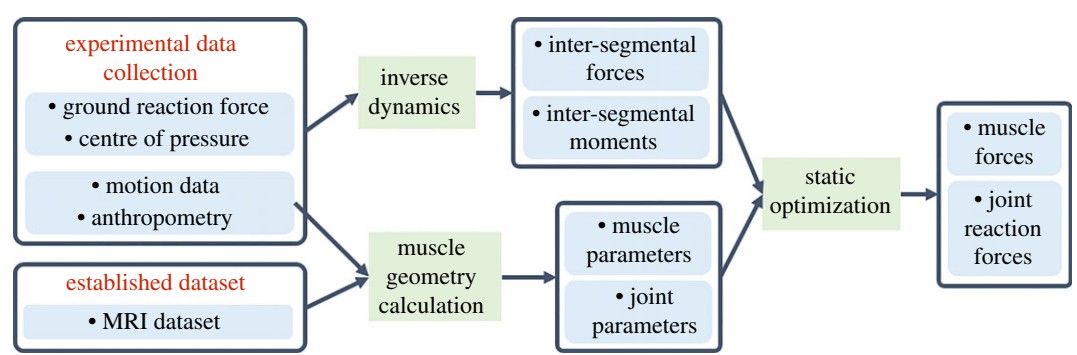

**Figure 2.** Modelling process.

the maximum current of 60 mA (1 kΩ). The frequency of the pulse was set to 40 Hz as recommended by the NMES manufacturer (Odstock 2 Channel Stimulator, Odstock Medical Ltd, UK). The pulse width was adjusted from 132.5 to 306.5 µs to generate visible muscle contraction at the limit of the subjects' tolerance. The stimulation was started manually by the subject by pressing a hand-held switch during the left foot strike before stepping on the force plate. The stimulation lasted 2 s to cover the entire stance phase of the right limb.

Right lower limb motion was recorded at 200 Hz using a nine-camera VICON motion analysis system (Vicon Motion Systems Ltd, Oxford, UK). Eighteen reflective markers were used in this study: RASIS, LASIS (right and left anterior superior iliac spine); RPSIS, LPSIS (right and left posterior superior iliac spine); T1, T2, T3 (cluster markers on the thigh); FLE, FME (lateral and medial femoral epicondyle); C1, C2, C3 (cluster markers on the shank); FAM, TAM (apex of the lateral and medial malleolus); FCC (calcaneus); FMT (tuberosity of the fifth metatarsal); FM2 (head of the second metatarsal) and TF (the centre of the acrotarsium). The system was calibrated using a 5 marker L frame.

GRFs were recorded at 1000 Hz from a force plate (Kistler Type 9286BA, Kistler Instrument AG, Winterthur, Switzerland). During stimulated walking, only the right leg muscles were stimulated, and the force plate recorded the force applied by the right foot only.

## 2.2. Modelling

The flowchart (figure 2) outlines the data flow and modelling process.

The open source musculoskeletal model Freebody 2.0, which is based on the original model of Cleather & Bull [33], was used [23]. The validation of this original model has been described previously [23,33–36] through comparison between the model calculations and *in vivo* measurements of joint reaction forces or EMG. The revised model with NMES has been validated [36] through the measurement of gluteus maximus activation using EMG pre- and post-application of NMES on loBF;

the kinematics and kinetics input to the model were measured simultaneously with the EMG, and strong positive correlations were found in early stance peak ($R = 0.78$, $p = 0.002$) and impulse ($R = 0.63$, $p = 0.021$). The original anatomical model used is from Klein *et al.* [37] with 163 muscle elements. However, in order to improve the relevance of the dataset to a clinical population, and improve on the joint contact point data, the anatomical geometry were newly derived from MR imaging data of one younger male (height: 1.83 m; mass: 96 kg; age: 43) selected from a database of anatomical models [38]. The anatomical geometry included the muscle origins, via points, and insertion points, along with joint centres of rotation, and contact points between the femur and tibial plateau.

In the local coordinate system of each segment (foot, shank, thigh and pelvis), the *x*-axis is in the anterior–posterior direction with anterior being positive; the *z*-axis is in the medial–lateral direction with lateral being positive and the *y*-axis is in the longitudinal direction of the segment (figure 1*b*). The anatomical model consists of the muscle insertion points of 163 muscle elements. These are scaled to each subject using scaling factors, where a muscle point in the anatomical model has coordinates $(x_0, y_0, z_0)$; this is multiplied by a scaling factor to obtain the coordinates of the muscle point in the subject $(x, y, z)$ (equation (2.1)). Scaling factors, $sf$, of the segments (pelvis, thigh, shank and foot) were determined from segmental ratios as in equation (2.2)

$$(x, y, z) = (x_0, y_0, z_0) \cdot sf \tag{2.1}$$

and

$$\left. \begin{aligned} sf_{\text{Pelvis}} &= \frac{\text{Pelvis width of subject}}{\text{Pelvis width in anatomical model}} \\ sf_{\text{Thigh}} &= \frac{\text{Thigh length of subject}}{\text{Thigh length in anatomical model}} \\ sf_{\text{Shank}} &= \frac{\text{Shank length of subject}}{\text{Shank length in anatomical model}} \\ sf_{\text{Foot}} &= \frac{\text{Foot length of subject}}{\text{Foot length in anatomical model}}. \end{aligned} \right\} \tag{2.2}$$

The model has five segments in total: foot, shank, thigh, pelvis and patella. A segment-based inverse dynamics calculation was conducted to quantify the intersegmental forces and moments [33], and an optimization-based approach was used to quantify the muscle forces [28,34]. The inverse dynamics method of Dumas *et al.* [39] was used to formulate the Newton–Euler equations of motion that include the segmental weight, external forces, muscle forces and joint reaction forces. The muscle forces were constrained to be between 0 and a maximal force, which was calculated by multiplying the physiological cross-sectional area (PCSA) from Klein's research by an assumed maximum muscle stress of $3.139 \times 10^5 \, \text{N m}^{-2}$ [37]. Muscle forces of 163 muscle elements, one patella ligament force and five joint reaction forces (ankle, medial knee, lateral knee, hip and patellofemoral) were considered.

The cost function of the static optimization is the sum of the cubic relative muscle forces, which was proposed based on the force–endurance relationship [32] (below equation)

$$\sum_{j=1}^{163} \left( \frac{F_j}{F_{j\max}} \right)^3, \tag{2.3}$$

where $F_j$ is the muscle force of muscle element $j$, and $F_{j\max}$ is the maximal muscle force of muscle element $j$.

The joint reaction forces except the tibiofemoral forces were considered to act through the centres of rotation of the joints taken from MR imaging data: the hip joint centre was defined as the centre of the femoral head, which was determined by fitting a sphere to the femoral head; the knee joint centre was defined at the midpoint of the central axis of a cylinder that was fitted to the boundaries of the femoral condyles; and the ankle joint centre was defined at the centre of a sphere that was fitted to the talar dome. The centres of rotation were fixed to the distal segment once determined. However, as the tibiofemoral joint reaction force is compartmentalized into a medial and a lateral component, the contact points of these two compartments are different from the knee joint centre of rotation, and they were located according to the MR imaging data of the subject mentioned above. The contact points were fixed to the shank segment and defined as the points of action of the medial and lateral knee joint forces; these were the middle of the most distal ends of the femoral condyles in the axial slice. These positions will change on the thigh segment according to the translations between two adjacent segments. The subjects' contact points were obtained by scaling these two points in the MR imaging data to the subject's shank using the shank scaling factor.

The outputs of the optimization are: magnitude of 163 muscle forces, one patella ligament force and five vectors of joint reaction forces for the ankle, lateral knee, medial knee, hip and patellofemoral joints.

## 2.3. NMES simulation

The optimization described above quantifies muscle forces according to an endurance function [32]. However, stimulated muscles are expected to have larger forces than in the normal case. In order to adapt this model to NMES-assisted walking, a new parameter $c$ was added to obtain the new cost function as in the below equation [40]

$$\sum_{j=1}^{163} c \cdot \left( \frac{F_j}{F_{j\max}} \right)^3 ,$$  (2.4)

where

$$c = \begin{cases} c_s & \text{for stimulated muscles} \\ 1 & \text{others.} \end{cases}$$  (2.5)

The adjustable weighting factor, $c_s$, enables the stimulated muscle force to increase relative to standard simulation for a value of $c_s$ of less than 1. Conversely, increasing $c_s$ to greater than 1 will result in under-activation of the muscle. As electrical stimulation recruits muscles directly, the combination of voluntary contractions with electrical stimulation produces stronger contraction [41] and so, clinically, $c_s$ is related to stimulation intensity and it is used to produce changes in muscle forces. In order to explore the sensitivity of $c_s$ on the muscle and joint forces, varying values of $c_s$ (1, 0.75, 0.5, 0.25, 0.1 and 0) were implemented. Finally, $c_s = 0.1$ for BF and $c_s = 0.25$ for latGAS and VL were selected to produce a significant increase in muscle force during the NMES simulation.

The new cost function (NMESsim) drives the optimization to produce larger forces for the stimulated muscles, and thus allows the simulation of NMES-assisted walking.

The joint reaction forces in normal gait were calculated with the original cost function in equation (2.3). As the imposition of muscle contraction using NMES could potentially alter the gait kinematics and kinetics, the effect of NMES on these variables only was assessed first using the original cost function. To investigate the effect of increasing muscle activation of the stimulated muscles, NMESsim in equations (2.4) and (2.5) was also used to quantify NMES-assisted walking. Joint kinematics were calculated as Euler angles based on the local coordinate systems of the adjacent segments [39]. The angles were referenced to the standing posture at which the joint angles were defined to be zero. The sequence of angle calculation is first flexion/extension (plantar flexion/dorsiflexion for the ankle), then external/internal rotation and lastly abduction/adduction.

## 2.4. Statistical analysis

The Wilcoxon signed-rank test (Matlab, R2014a, The Mathworks, Inc., 2014) was used to determine if there was a significant difference between normal walking and NMES-assisted walking. Muscle forces were compared to validate the effectiveness of NMESsim. As two different cost functions were used, and three muscles were stimulated, the loBF, latGAS and VL muscle forces in normal gait were compared with those forces during stimulated walking using both the original cost function and NMESsim.

The magnitude (in body weight—BW) of KMF and KAM were compared between cases. The ankle, knee and hip joint angles were calculated and compared for normal and NMES-stimulated walking tasks.

The two peak values of GRF and KFM, mean walking velocity and step length during the stance phase were compared between normal gait and NMES-stimulated walking. In all analyses, each of the three trials per subject were analysed, and then mean parameters are presented.

Anonymized data files from the physical experiments and the computational simulations are available on request from the corresponding author.

# 3. Results

## 3.1. Sensitivity analysis

The muscle forces and KMFs with varying $c_s$ for eight subjects are plotted in figure 3. The stimulated muscle forces (including loBF, latGAS and VL forces) increase from the turquoise line ($c_s = 1$) to the

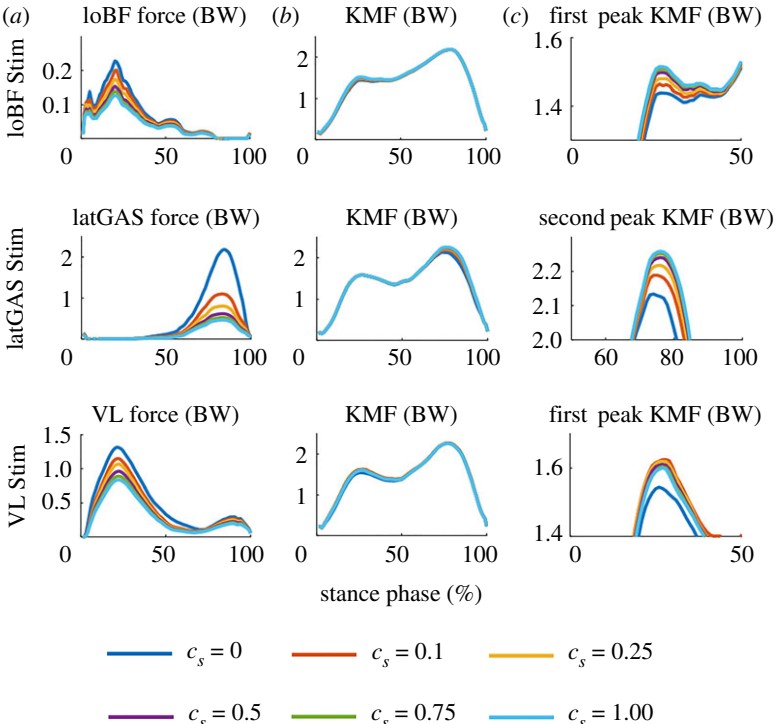

**Figure 3.** ($a-c$) Muscle forces and KMFs in BW for varying $c_s$. The graphs in column $c$ represent enlarged portions of the graphs in column $b$.

blue line ($c_s = 0$) as $c_s$ decreases (figure 3$a$). The first peak of KMF decreases as the loBF force increases, and the second peak of KMF decreases as the latGAS force increases, from the turquoise line to the blue line. The VL has a similar effect as loBF on the first peak, but the decrease in KMF only became clearly apparent with $c_s = 0$.

## 3.2. Muscle forces

Figure 4 depicts the mean forces of the main lower limb muscles of eight subjects using NMESsim during four tasks. The nine muscles are loBF, latGAS, VL, medial head of GAS (medGAS), soleus (SL), tibialis anterior (TA), rectus femoris (RF), gluteus maximus (GlutMax) and gluteus medius (GlutMed).

There was no significant difference between maximal muscle force values in normal gait and stimulated walking when using the original cost function. However, maximal muscle forces in the stimulated tasks were significantly larger than the maximal muscle forces in normal gait when using NEMSsim for stimulated walking (loBF: 0.27 BW versus 0.20 BW, $p = 0.039$; latGAS: 0.61 BW versus 0.34 BW, $p = 0.008$; VL: 1.18 BW versus 0.78 BW, $p = 0.008$; table 1).

## 3.3. Knee forces and moments

The mean KMFs are shown in figure 5. Generally, the KMF has two peaks during the stance phase occurring at the mean values of 28 and 78% of the stance phase. However, two subjects do not have two explicit peaks. Therefore, the mean KMFs of 20–35 and 70–85% of the stance phase are calculated to represent the two peak values. The second peak KMF in loBF-stimulated walking is decreased significantly using the original cost function (stimulated versus normal: 2.26 BW versus 2.32 BW, $p = 0.039$) and NMESsim (stimulated versus normal: 2.15 BW versus 2.32 BW, $p = 0.016$) in table 2. The peak KMFs of each subject are shown in table 3.

The mean KAMs and KFMs are shown in figure 6. The second peak of KAM during loBF-stimulated walking is significantly smaller than that during normal walking (1.69% BW × Ht versus 1.85% BW × Ht, $p = 0.016$, table 4). However, peak KFM does not change with NMES.

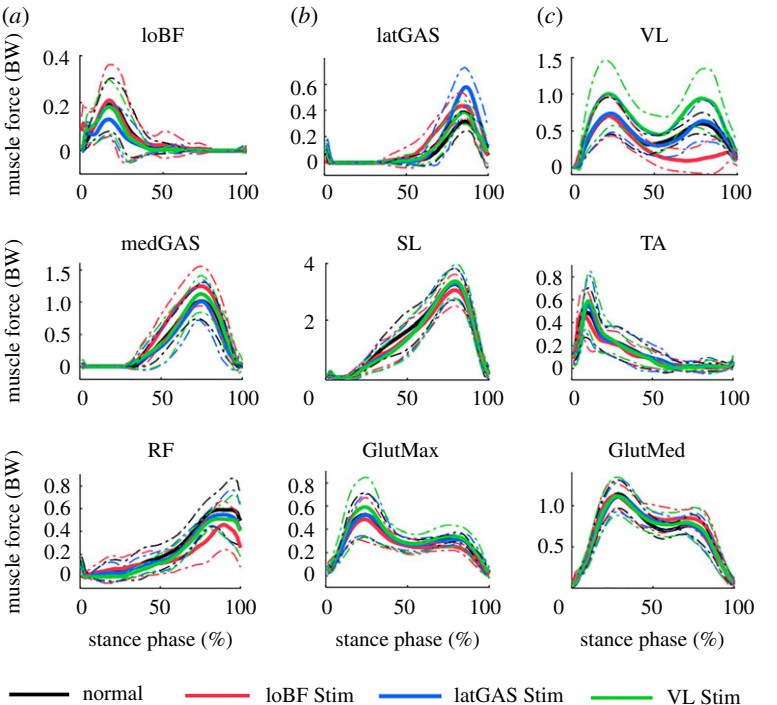

**Figure 4.** Muscle forces obtained from NMESsim ($n = 8$). Solid line, mean values; dashed line, standard deviations. Normal, normal walking; loBF Stim, loBF-stimulated walking; latGAS Stim, latGAS-stimulated walking; VL Stim, VL-stimulated walking.

**Table 1.** Maximal muscle forces in BW of stimulated (NMESsim) and normal (original cost function) walking (mean $\pm$ standard deviation). The maximal muscle force in italics indicates that it is significantly different from the maximal force of the same muscle in normal walking.

| muscle | normal walking | loBF-stimulated case | latGAS-stimulated case | VL-stimulated case |
|--------|----------------|----------------------|------------------------|---------------------|
| loBF | 0.20 ± 0.11 | *0.27 ± 0.16** | 0.15 ± 0.08 | 0.20 ± 0.12 |
| latGAS | 0.34 ± 0.08 | *0.45 ± 0.10*** | *0.61 ± 0.14**** | 0.39 ± 0.09 |
| VL | 0.78 ± 0.20 | 0.72 ± 0.28 | 0.81 ± 0.23 | *1.18 ± 0.38**** |

\*$p = 0.039$, \*\*$p = 0.016$, \*\*\*$p = 0.008$.

## 3.4. Ground reaction forces, walking velocity and step length

GRF is presented in figure 7. There was no statistical difference in peak GRFs, walking velocity and step length between NMES stimulated and normal walking (table 5). The peak GRFs were defined as those at the same timing as the peak KMFs. Specifically, these were calculated as the mean values across 20–35 and 70–85% of the stance phase.

## 4. Discussion

This study has shown that the musculoskeletal system can be tuned by increasing muscle forces to reduce medial knee joint contact forces (decreased peak values of KMF) in healthy subjects through the excitation of loBF, and has tested this using NMES. This opens up the possibilities of using NMES to treat knee OA, a major clinical condition causing disability and pain.

This study is not in agreement with previous computational studies that found that no muscles that cross the knee could reduce the KMF [24,42]. A key difference between these studies and the present study is that they assumed that the muscle activations were calculated without kinematic compensations, i.e. the kinematic and kinetic datasets remained the same for normal and stimulated

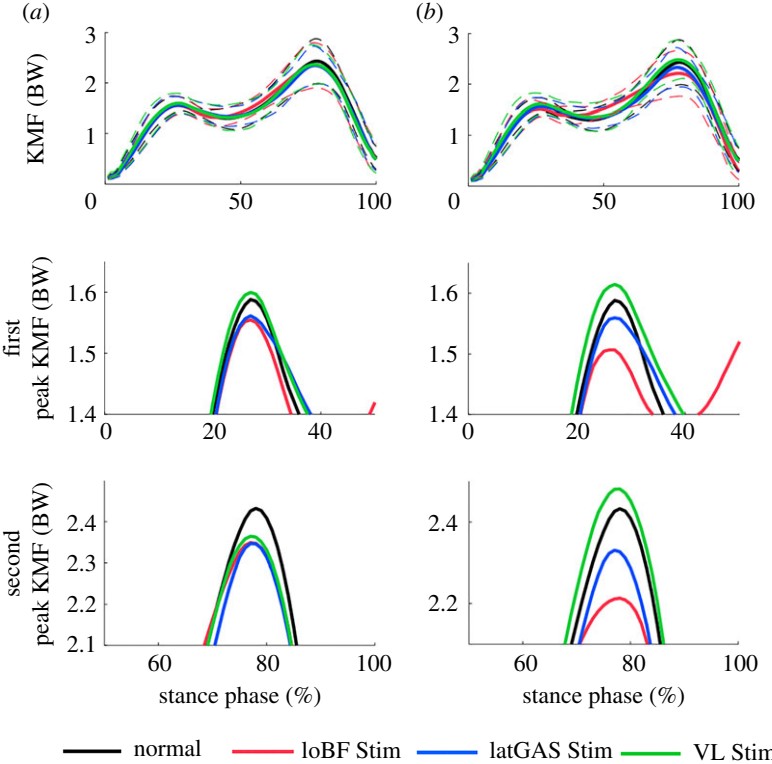

**Figure 5.** Mean KMF obtained using the original cost function (*a*) and NMESsim (*b*).

**Table 2.** Peak KMFs in BW (mean ± standard deviation). The peak knee medial force in italics indicates that it is significantly different from that of normal gait.

| task | parameter | method | |
|---|---|---|---|
| | | original cost function | NMESsim |
| normal | first peak | 1.51 ± 0.14 | — |
| | second peak | 2.32 ± 0.39 | — |
| loBF Stim | first peak | 1.48 ± 0.17 | 1.45 ± 0.16 |
| | second peak | *2.26 ± 0.41** | *2.15 ± 0.43*** |
| latGAS Stim | first peak | 1.50 ± 0.17 | 1.50 ± 0.17 |
| | second peak | 2.24 ± 0.31 | 2.22 ± 0.32 |
| VL Stim | first peak | 1.53 ± 0.18 | 1.55 ± 0.20 |
| | second peak | 2.27 ± 0.33 | 2.38 ± 0.32 |

$*p = 0.039$, $**p = 0.016$.

walking. They did not stimulate the muscles, and did not incorporate the changing kinematics and kinetics due to NMES. One of the advantages of the present study is that the kinematics and kinetics caused by increased muscle activations were used in the calculations. The different results of these studies highlight the importance of quantifying the effect of stimulation on kinematics and kinetics.

## 4.1. The use of $c_s$

Voluntary contractions preferentially recruit type I fibres, and then type II fibres. This asynchronous activation of varied motor units is a physiological mechanism that decreases muscle fatigue [41]. However, electrically elicited contractions recruit motor units synchronously, and recruit motor units

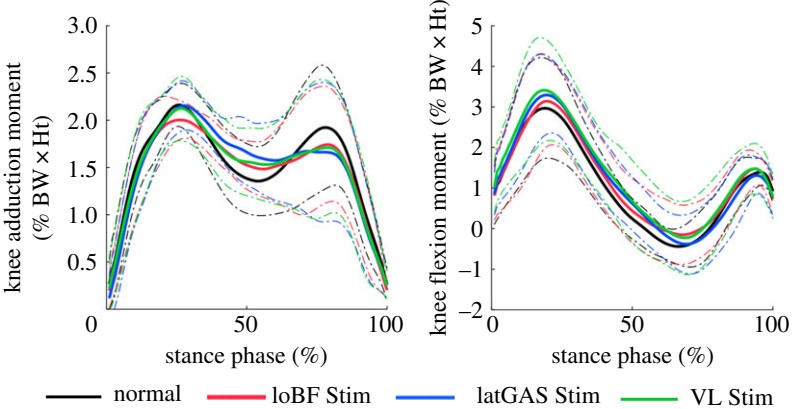

**Figure 6.** Knee adduction moments. Solid line, mean values; dashed line, standard deviations. BW × Ht, body weight × height.

**Table 3.** Peak KMF of each subject in BW. The values in italics are smaller than the 'normal' values.

| parameter | subject | normal | loBF Stim | latGAS Stim | VL Stim |
|---|---|---|---|---|---|
| first peak | 1 | 1.42 | *1.29* | *1.37* | *1.36* |
| | 2 | 1.65 | *1.50* | *1.56* | 1.74 |
| | 3 | 1.52 | 1.59 | 1.64 | 1.70 |
| | 4 | 1.39 | *1.27* | *1.32* | *1.37* |
| | 5 | 1.50 | *1.39* | *1.41* | *1.39* |
| | 6 | 1.75 | *1.67* | 1.80 | 1.84 |
| | 7 | 1.53 | 1.62 | 1.58 | 1.64 |
| | 8 | 1.34 | *1.31* | 1.34 | 1.38 |
| second peak | 1 | 2.21 | 2.28 | *2.19* | 2.37 |
| | 2 | 1.95 | *1.70* | *1.76* | 1.97 |
| | 3 | 1.86 | *1.76* | 1.87 | 2.18 |
| | 4 | 2.10 | *1.75* | 2.16 | *2.09* |
| | 5 | 2.69 | *2.43* | *2.54* | 2.82 |
| | 6 | 2.68 | *2.51* | *2.40* | 2.72 |
| | 7 | 2.93 | *2.85* | *2.70* | *2.70* |
| | 8 | 2.14 | *1.91* | *2.12* | 2.21 |

that are not activated by voluntary contractions [41]. It is deduced that stimulated muscles produce larger forces than voluntary conditions, and the other muscles contract in a way to decrease muscle fatigue. In view of this, NMESsim was used with a parameter $c$ to realize a reasonable muscle force distribution: the stimulated muscle force was increased, and all the muscle and joint forces satisfied the motor function constraints. The $c_s$ values were chosen between 0 and 1 (0, 0.1, 0.25, 0.5, 0.75 and 1), where $c_s = 1$ represents the original cost function. Where the muscles are not saturated (i.e. limited by their PCSA), assigning a value of $c_s$ of less than 1 enforces a greater muscle force than when using the original cost function; a value of $c_s = 0$ assigns the greatest activation to the muscle as this removes the muscle from the cost function.

Using $c_s = 0.1$ for BF and $c_s = 0.25$ for latGAS and VL increased the activation of the stimulated muscles (table 1), but did not, in this study, saturate the muscle (one of the constraints requires the muscle force to be less than the maximal muscle force). Therefore, theoretically, the simulated muscle force with $c_s = 0.1$ and $c_s = 0.25$ is expected to be practically achievable. The physical experiment used NMES to contract the muscles at the maximum comfortable level for each subject.

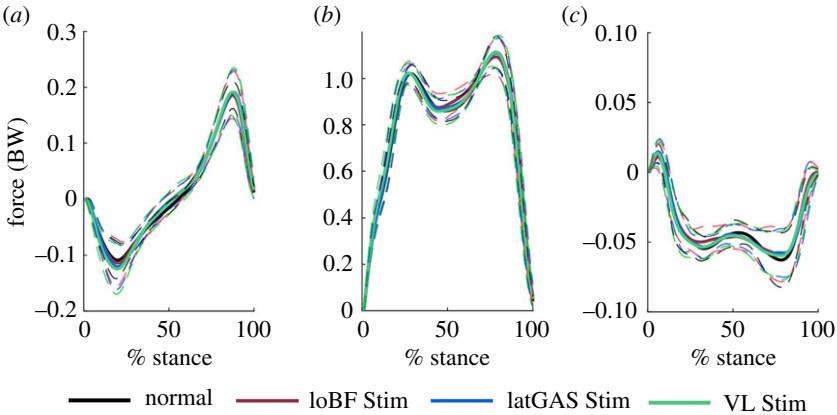

**Figure 7.** Components of GRF in the forward/backward direction (GRFx, a); upward/downward direction (GRFy, b) and right/left direction (GRFz, c).

**Table 4.** KAM and KFM in % BW × HT (mean ± standard deviation). The moment in italics indicates that it is significantly different from that of normal gait.

|  | task | first peak | second peak |
| --- | --- | --- | --- |
| KAM | normal | 2.07 ± 0.24 | 1.85 ± 0.60 |
|  | loBF Stim | 1.96 ± 0.20 | *1.69 ± 0.59*\* |
|  | latGAS Stim | 2.08 ± 0.25 | 1.64 ± 0.68 |
|  | VL Stim | 2.04 ± 0.31 | 1.67 ± 0.67 |
| KFM | normal | 2.42 ± 0.97 | 0.09 ± 0.62 |
|  | loBF Stim | 2.62 ± 0.91 | 0.24 ± 0.68 |
|  | latGAS Stim | 2.74 ± 0.84 | 0.00 ± 0.77 |
|  | VL Stim | 2.80 ± 1.00 | 0.25 ± 0.89 |

\*$p = 0.016$.

**Table 5.** GRF, walking velocity and step length (mean ± standard deviation).

|  |  | normal | loBF Stim | latGAS Stim | VL Stim |
| --- | --- | --- | --- | --- | --- |
| GRFx (BW) | first peak | −0.09 ± 0.02 | −0.09 ± 0.02 | −0.09 ± 0.02 | −0.09 ± 0.03 |
|  | second peak | 0.12 ± 0.02 | 0.12 ± 0.03 | 0.12 ± 0.03 | 0.12 ± 0.02 |
| GRFy (BW) | first peak | 0.97 ± 0.02 | 0.98 ± 0.04 | 0.98 ± 0.03 | 0.98 ± 0.04 |
|  | second peak | 1.08 ± 0.06 | 1.06 ± 0.06 | 1.08 ± 0.05 | 1.08 ± 0.06 |
| GRFz (BW) | first peak | −0.05 ± 0.01 | −0.05 ± 0.01 | −0.05 ± 0.01 | −0.05 ± 0.01 |
|  | second peak | −0.06 ± 0.02 | −0.06 ± 0.02 | −0.06 ± 0.02 | −0.06 ± 0.02 |
| walking velocity (m s$^{-1}$) |  | 0.78 ± 0.08 | 0.71 ± 0.13 | 0.75 ± 0.13 | 0.77 ± 0.14 |
| step length (m) |  | 0.58 ± 0.05 | 0.57 ± 0.06 | 0.57 ± 0.06 | 0.58 ± 0.06 |

The new algorithm, NMESsim, succeeded in simulating the stimulated condition, because the model-predicted muscle forces under stimulation were significantly larger than those of normal cases and all the muscle forces and joint reaction forces satisfied the equations of motion.

EMG signals are affected by the stimulation artefacts caused by the stimulation current when NMES is used. This has been partially addressed in the literature with, for example, use of an EMG-amplifier with shut-down control [43], advanced filtering procedures [44] and optimizing the positioning of the EMG electrodes in relation to the stimulation electrodes [45]. However, these approaches do not fully

eliminate the stimulation artefacts and we, therefore, were unable to use EMG as a validation of the method.

## 4.2. KMF and KAM

The shape of the KMF in stance phase has an expected double hump pattern with slightly higher peak than measured *in vivo* [19–21,26,46,47]. The literature has much information on the reasons for higher joint force predictions using musculoskeletal modelling; in this case, we hypothesize that such forces are appropriate due to our subjects being younger and more active than those subjects in the literature who had total joint replacements.

The decrease in KMF due to NMES is not substantial compared to the reduction caused by other methods. Kinney *et al.* [47] quantified KMF reduction through gait modification using long hiking poles with wide pole placement, and found that KMF at 75% stance decreased significantly from 1.35 BW to 0.89 BW. Other modifications in Kinney's study, such as mild crouch and medial thrust, did not show any significant KMF reduction. Orthotic interventions using a lateral foot wedge and valgus knee brace resulted in a decrease in model-estimated peak tibiofemoral load with increasing wedge size and applied valgus brace moment [7]. A higher level of activation using NMES might be required to create a greater reduction in KMF.

Although some of variables were not significantly different between groups, this does not negate the fact that changes can be significant for an individual. Table 3 lists how peak KMF varies for eight subjects and although the KMF decrease for the group is not substantial, the simulation could be used to identify which patient is well suited to this intervention.

The KAM curves shown in figure 6 are similar to those of Manal *et al.* [21]. The KAM is reported to be highly correlated with the medial contact force for different gait patterns, and is thus often used as a surrogate measure [19]. The significant decrease in KAM by loBF stimulation was consistent with that of KMF. A study of 62 OA patients verified the clinical importance of KAM in that peak KAM and KAM impulse explained variance in knee cartilage thickness [48]. However, whether the change of KAM could contribute to the alleviation of OA symptoms needs clinical validation.

## 4.3. GRF, walking velocity and step length

Changes in walking velocity and step length have been found to affect knee joint contact forces [49]. The GRF, as the external force on the whole body, is related to the overall kinematics and kinetics. Here, there is no significant difference in GRF, velocity and step length, so although the NMES changed the subtle kinematics of the lower limb, the lack of overall changes in pace could explain the lack of variation in GRF.

Although normal gait is not perfectly symmetrical, stimulation of one side alone makes the gait even more asymmetrical. However, the results here show that these changes were small and not large enough to obviously disturb normal walking as the GRF, walking velocity and step length of stimulated walking are similar to that of normal walking.

## 4.4. Limitations and future work

This study simulated the situations where only one muscle was stimulated. However, it is possible that the stimulation of combined muscles will not change the net joint moment, but together have a greater impact on KMF than stimulating each muscle independently. This case should be considered further.

The muscle forces of loBF, latGAS and VL generate movement of the lower limb mainly through changing the flexion–extension moments of the knee and ankle. The extra muscle activation by NMES may influence the periodic characteristics of normal gait. In this way, different timing of NMES leads to different gait characteristics, and will affect the reduction in KMF. Therefore, NMES timing is an important parameter for precise control of NMES. This study used a very crude manually implemented timing mechanism that should be tuned for optimal outcome.

The decrease in the KMF was limited by the increase in the muscle force, which was determined by the $c_s$ value in the model. Lower $c_s$ values than were applied in this study could be used to generate a greater reduction in KMF. However, the clinical feasibility of including such high muscle forces is not known.

This study is a pilot study on the effect of NMES on the KMF, and only healthy subjects were considered. The study verified that the stimulation of loBF will redistribute the knee loading by

decreasing the KMF. The result provides a new way to control the knee loading distribution. These results suggest that it may also be possible to decrease the medial knee joint loading of OA patients using NMES. Therefore, the next steps in this work should involve the recruitment of OA patients in order to verify the effect of NMES on a pathological group.

# 5. Conclusion

A new cost function for use in musculoskeletal models, NMESsim, has been proposed to allow the quantification of muscle forces when they are stimulated by external means. This was tested in living subjects, and our results show that the stimulated muscle forces are significantly increased (loBF: $\Delta = 0.07$ BW, $p = 0.039$; latGAS: $\Delta = 0.27$ BW, $p = 0.008$; VL: $\Delta = 0.40$ BW, $p = 0.008$) and the peak values of knee joint medial loading are significantly decreased by applying NMES to loBF (NMESsim: $\Delta = 0.17$ BW, $p = 0.016$). This study demonstrates that it is possible to redistribute the knee loading and reduce the loading on the medial compartment by activating selected muscles across the healthy knee and this opens the door for prevention and alternative non-surgical interventions for knee OA.

Ethics. The experiment was approved by the ethics committee of Imperial College London (reference number: 14IC2134), and all subjects gave written informed consent.

Data accessibility. Anonymized experimental data used in this study can be found at DOI Identifier: https://doi.org/10.5281/zenodo.2579677. The musculoskeletal model Freebody can be accessed from http://www.msksoftware.org.uk.

Authors' contributions. All authors contributed to conception and design of the study and were involved in drafting and critically revising the manuscript. Additionally, R.X. carried out the experimental work, mathematical modelling, interpreted the results and prepared the first draft paper. Z.D. carried out mathematical modelling. A.M.J.B. provided interpretation of the experimental and modelling results and worked up the draft paper into the final version. All authors gave final approval for publication.

Competing interests. The authors have no competing interests.

Funding. This research was supported by the Medical Engineering Solutions in Osteoarthritis Centre of Excellence at Imperial College London, which is funded by the Wellcome Trust and the Engineering and Physical Sciences Research Council, UK. R.X. was supported by the China Scholarship Council.

Acknowledgements. The authors would like to thank all the participants of the experiment, especially the Musculoskeletal Biomechanics Group, Imperial College London.

# Appendix A

The normalized force–endurance relationship formula is

$$\log T = -\log\left(\frac{f}{f_{\max}}\right)^n + c, \qquad (A\,1)$$

where $T$ is the endurance time, $c$ is a constant, $f$ is the muscle force, $f_{\max}$ is the maximal muscle force and $n$ usually ranges from 2.54 to 3.14. The equation illustrates that the endurance time increases as the relative muscle force $(f/f_{\max})^n$ decreases.

The muscle selection to maximize activity endurance is physiologically reasonable during many normal activities. Therefore, the musculoskeletal model to mimic muscle contraction for normal activities has to maximize muscle endurance, which is realized by minimizing $\Sigma(f_i/f_{i\max})^3$, i.e. the sum of $(f_i/f_{i\max})^3$ for all muscles.

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
