## [Reviewer comments · Royal Society Open Science]

Review History

RSOS-171490.R0 (Original submission)

Review form: Reviewer 1

Is the manuscript scientifically sound in its present form?

No

Are the interpretations and conclusions justified by the results?

No

Is the language acceptable?

No

Is it clear how to access all supporting data?

Not Applicable

Do you have any ethical concerns with this paper?

No

Have you any concerns about statistical analyses in this paper?

Yes

Recommendation?

Reject

Comments to the Author(s)

The manuscript is poor in English in the current format. In many cases the present tense and past tense were mixed.

There are quite some unclarified issues in the models used, for example the meaning of Cs in clinical application, how the KMF acting point was identified? was it varied like the COP of GRF during stance?

Cs was not clarified in the application. Did it have a specific clinical meaning or just a variable used to produce changes in the results.

The standard deviation of some primary results are too high, the authors didn't present the GRF and the walking speed, which made it hard to assess the quality of the data. Ideally a test and retest reliability should have been performed to assure the data quality.

Some results are hard to comprehend, such as the knee medial force is directly proportional to the knee adduction moment, however the manuscript presented data curves that showed some contrary results, In the KMF curves the second peak was much higher than the first peak, however in the curves of KAM the first peak was much higher than the second. It is very hard to understand the results as the force should be directly proportional to the knee adduction moment.

Additionally, the KMF and KAM should have the similar variation trend, for example in the KAM curves the KAM of the normal condition was the highest 2nd peak, the others were much lower than it. Contrarily the KMF curves showed the VL Stim was the highest in the second peak, which was hard for readers to understand.

Review form: Reviewer 2**Is the manuscript scientifically sound in its present form?**

No

Are the interpretations and conclusions justified by the results?

No

Is the language acceptable?

Yes

Is it clear how to access all supporting data?

Yes

Do you have any ethical concerns with this paper?

No

Have you any concerns about statistical analyses in this paper?

No

Recommendation?

Reject

Comments to the Author(s)

General Comments

This paper presents an interesting idea, to alter the muscle force distribution using NMES during walking as a potential intervention for medial tibiofemoral OA. However, there are a number of concerns that limit the impact of this work and the conclusions that can be drawn. These concerns are highlighted below.

The authors assume the use of the knee adduction moment (KAM), as a surrogate measure of the medial force in the knee, as many others also do. However, the KAM is one of 6 inverse dynamic loads and is not necessarily a good predictor of the medial contact force in the knee (KMF), as it does not account for muscle activity across the joint. In the context of this paper, where you are altering muscle forces using NMES would you expect to see a change in the KAM?

The addition of the parameter, 'c', seems to have the desired effect of changing the muscle force distribution. However, the selection of the parameter value seems a little arbitrary. How is this physiologically related to the relative increase in muscle force from NMES? Can this be estimated by measurement of EMG, or other methods?

The changes in the KAM presented in Figure 7 show very large standard deviations and it is interesting that there are statistically significant differences in the 2nd peak of the KAM between the Normal and stim conditions. Although there might be statistical significance, these differences might not be clinically meaningful. The same can be said about the change in ankle adduction angle. A difference of 0.4 deg between the Normal and stim condition does not seem to be clinically meaningful, nor does it appear to be within the accuracy of the measurement system.

The authors conclude that NMES provides an opportunity to treat knee osteoarthritis. The changes in KMF due to the stim condition was considered 'not substantial' (page 12, line 57), when compared to some other forms of gait adaptation or gait retraining in the literature. It would be useful to have this discussion in the context of these other modifications. Furthermore, the large variability seen in the KAM results (which reflects the variability you might expect across a patient cohort), demonstrates that perhaps this intervention would work for some individuals, but not others. It would be nice if the simulation could identify which patient might be well-suited to this intervention.

There is no discussion regarding the potential changes in the flexion-extension moments following stimulation of each muscle. I also anticipated some discussion regarding the stimulation of combined muscles, which might not change the net joint moment, but together might have a greater impact on KMF than stimulating each muscle independently. The authors conclude that the stimulated muscle forces were significantly increased and the KMF was significantly decreased. Without stating the magnitude of these changes, a reader is left with the opinion that these differences were considerable, which is misleading. Furthermore, the evidence to support these statements is not entirely convincing. Firstly, the findings presented regarding muscle force were taken from a model in which the cost function was altered to achieve a desired result of redistributing the muscle force. Secondly, the authors previously state in the discussion that the resulting KMF from the stim conditions were 'not substantial'.

Specific Comments

1. Abstract. It would be good to have some quantitative measure of the amount of change in the peak knee medial force, rather than just stating the difference was statistically significant. What was the absolute and/or percentage change? The authors state that there was a strong correlation between knee medial force and knee adduction moment, but again, it would be useful to illustrate the strength of this correlation by supporting the statement with evidence (i.e. an $R^2 =$

0.9).

2. Page 3. Line 5. The authors cite a few papers from their own research group here, but this does not really reflect the large efforts that are ongoing amongst the computational biomechanics community to use musculoskeletal modelling to estimate internal forces.

3. Page 3 line 11. In vivo measurement of knee contact forces suggests that there is rather poor evidence of the efficacy of valgus bracing and lateral wedges to reduce knee loading (Kutzner, I., Küther, S., Heinlein, B., Dymke, J., Bender, A., Halder, A. M., & Bergmann, G. (2011). The effect of valgus braces on medial compartment load of the knee joint - in vivo load measurements in three subjects. *Journal of Biomechanics*, 44(7), 1354–1360; and Kutzner, I., Damm, P., Heinlein, B., Dymke, J., Graichen, F., & Bergmann, G. (2011). The effect of laterally wedged shoes on the loading of the medial knee compartment-in vivo measurements with instrumented knee implants. *Journal of Orthopaedic Research*, 29(12), 1910–1915.). The efficacy of such devices to influence clinical symptoms might be different, but the evidence provided by the authors to claim that there is good support for reducing medial tibiofemoral joint contact forces is not 'firm'.

4. Page 4. Line 18. The authors use computational models to estimate the knee force, which is not a 'direct estimation', but rather an indirect one.

5. Page 4 Line 21. It is not clear how the use of the knee adduction moment (KAM) improves the reliability of the results? The KAM provides an 'easy to measure' surrogate of the KMF, but only represents one of 6 inverse dynamic loads and does not account for muscle co-contraction. As such, the KAM is not necessarily a strong predictor of KMF.

6. Page 5. Line 10. It seems like the timing of the increased muscle activation might be important, given these muscles also play a role in developing flexion-extension moments? This might be worth mentioning.

7. Page 6. Line 37. It is unclear how Equation (1) shows a force-endurance relationship from Crowninshield and Brand (1981), as it merely implies a ratio of a maximum muscle force?

8. Page 7. Line 16. What are the 5 vectors of reaction forces? These should be explicitly stated here.

9. Page 8. Line 16. Suggest restructuring sentence.

10. Page 8. Line 22. This is an unusual Euler sequence of rotations. Typically you select the degree of freedom that has the greatest motion as the first rotation (i.e. flexion-extension). Doing otherwise might result in greater kinematic crosstalk.

11. Page 9. Paragraphs 2, 3 and 4. When describing results, the authors commonly state 'significantly larger', however, do not present any statistical evidence for these statements.

12. Page 10. Line 10. It is not clear whether the decreased ankle abduction observed in the experiment resulted directly in a reduction of medial knee contact force, as suggested. This finding represents an association and not a causation.

13. Page 13. Line 3. Using a lower value of c_s in the cost function resulted in an 'obvious reduction in KMF', however, how realistic or physiological would this be?

14. Page 13 line 29. The statement that the ankle adduction angle and KAM 'strengthened the reliability' of the KMF decrease due to latGAS stim is a hypothesis and should maybe be presented in discussion, not the conclusion. Compared to the 'Normal' condition, the change in adduction was only 0.4 degrees (with a SD of 2.1 degrees), which is likely below the accuracy level of the kinematic model.

Decision letter (RSOS-171490.R0)

20-Dec-2017

Dear Dr Xu:

Manuscript ID RSOS-171490 entitled "Extra Excitation of Biceps Femoris or Lateral

Gastrocnemius during NMES Reduces Knee Medial Loading" which you submitted to Royal Society Open Science, has been reviewed. The comments from reviewers are included at the bottom of this letter.

In view of the criticisms of the reviewers, the manuscript has been rejected in its current form. However, a new manuscript may be submitted which takes into consideration these comments.

Please note that resubmitting your manuscript does not guarantee eventual acceptance, and that your resubmission will be subject to peer review before a decision is made.

Your resubmitted manuscript should be submitted by 19-Jun-2018. If you are unable to submit by this date please contact the Editorial Office.

Please note that Royal Society Open Science will introduce article processing charges for all new submissions received from 1 January 2018. Charges will also apply to papers transferred to Royal Society Open Science from other Royal Society Publishing journals, as well as papers submitted as part of our collaboration with the Royal Society of Chemistry (<http://rsos.royalsocietypublishing.org/chemistry>). If your manuscript is submitted and accepted for publication after 1 Jan 2018, you will be asked to pay the article processing charge, unless you request a waiver and this is approved by Royal Society Publishing. You can find out more about the charges at <http://rsos.royalsocietypublishing.org/page/charges>. Should you have any queries, please contact openscience@royalsociety.org.

on behalf of Dr Michael Doube (Associate Editor) and R. Kerry Rowe (Subject Editor)
openscience@royalsociety.org

Associate Editor Comments to Author:

Dear Dr Xu,

Two reviewers have assessed your manuscript and find substantial weakness in the analysis and interpretation of your data. We will be willing to consider a resubmission of your manuscript if you are able to address all the reviewers' concerns.

Best regards,

Michael Doube, PhD

Reviewers' Comments to Author:

Reviewer: 1

Comments to the Author(s)

The manuscript is poor in English in the current format. In many cases the present tense and past tense were mixed.

There are quite some unclarified issues in the models used, for example the meaning of Cs in clinical application, how the KMF acting point was identified? was it varied like the COP of GRF during stance?

Cs was not clarified in the application. Did it have a specific clinical meaning or just a variable used to produce changes in the results.

The standard deviation of some primary results are too high, the authors didn't present the GRF and the walking speed, which made it hard to assess the quality of the data. Ideally a test and retest reliability should have been performed to assure the data quality.

Some results are hard to comprehend, such as the knee medial force is directly proportional to the knee adduction moment, however the manuscript presented data curves that showed some contrary results. In the KMF curves the second peak was much higher than the first peak, however in the curves of KAM the first peak was much higher than the second. It is very hard to understand the results as the force should be directly proportional to the knee adduction moment.

Additionally, the KMF and KAM should have the similar variation trend, for example in the KAM curves the KAM of the normal condition was the highest 2nd peak, the others were much lower than it. Contrarily the KMF curves showed the VL Stim was the highest in the second peak, which was hard for readers to understand.

Reviewer: 2

Comments to the Author(s)

General Comments

This paper presents an interesting idea, to alter the muscle force distribution using NMES during walking as a potential intervention for medial tibiofemoral OA. However, there are a number of concerns that limit the impact of this work and the conclusions that can be drawn. These concerns are highlighted below.

The authors assume the use of the knee adduction moment (KAM), as a surrogate measure of the medial force in the knee, as many others also do. However, the KAM is one of 6 inverse dynamic loads and is not necessarily a good predictor of the medial contact force in the knee (KMF), as it does not account for muscle activity across the joint. In the context of this paper, where you are altering muscle forces using NMES would you expect to see a change in the KAM?

The addition of the parameter, 'c', seems to have the desired effect of changing the muscle force distribution. However, the selection of the parameter value seems a little arbitrary. How is this physiologically related to the relative increase in muscle force from NMES? Can this be estimated by measurement of EMG, or other methods?

The changes in the KAM presented in Figure 7 show very large standard deviations and it is interesting that there are statistically significant differences in the 2nd peak of the KAM between the Normal and stim conditions. Although there might be statistical significance, these differences might not be clinically meaningful. The same can be said about the change in ankle adduction angle. A difference of 0.4 deg between the Normal and stim condition does not seem

to be clinically meaningful, nor does it appear to be within the accuracy of the measurement system.

The authors conclude that NMES provides an opportunity to treat knee osteoarthritis. The changes in KMF due to the stim condition was considered 'not substantial' (page 12, line 57), when compared to some other forms of gait adaptation or gait retraining in the literature. It would be useful to have this discussion in the context of these other modifications. Furthermore, the large variability seen in the KAM results (which reflects the variability you might expect across a patient cohort), demonstrates that perhaps this intervention would work for some individuals, but not others. It would be nice if the simulation could identify which patient might be well-suited to this intervention.

There is no discussion regarding the potential changes in the flexion-extension moments following stimulation of each muscle. I also anticipated some discussion regarding the stimulation of combined muscles, which might not change the net joint moment, but together might have a greater impact on KMF than stimulating each muscle independently. The authors conclude that the stimulated muscle forces were significantly increased and the KMF was significantly decreased. Without stating the magnitude of these changes, a reader is left with the opinion that these differences were considerable, which is misleading. Furthermore, the evidence to support these statements is not entirely convincing. Firstly, the findings presented regarding muscle force were taken from a model in which the cost function was altered to achieve a desired result of redistributing the muscle force. Secondly, the authors previously state in the discussion that the resulting KMF from the stim conditions were 'not substantial'.

Specific Comments

1. Abstract. It would be good to have some quantitative measure of the amount of change in the peak knee medial force, rather than just stating the difference was statistically significant. What was the absolute and/or percentage change? The authors state that there was a strong correlation between knee medial force and knee adduction moment, but again, it would be useful to illustrate the strength of this correlation by supporting the statement with evidence (i.e. an $R^2 = 0.9$).
2. Page 3. Line 5. The authors cite a few papers from their own research group here, but this does not really reflect the large efforts that are ongoing amongst the computational biomechanics community to use musculoskeletal modelling to estimate internal forces.
3. Page 3 line 11. In vivo measurement of knee contact forces suggests that there is rather poor evidence of the efficacy of valgus bracing and lateral wedges to reduce knee loading (Kutzner, I., Küther, S., Heinlein, B., Dymke, J., Bender, A., Halder, A. M., & Bergmann, G. (2011). The effect of valgus braces on medial compartment load of the knee joint - in vivo load measurements in three subjects. *Journal of Biomechanics*, 44(7), 1354-1360; and Kutzner, I., Damm, P., Heinlein, B., Dymke, J., Graichen, F., & Bergmann, G. (2011). The effect of laterally wedged shoes on the loading of the medial knee compartment-in vivo measurements with instrumented knee implants. *Journal of Orthopaedic Research*, 29(12), 1910-1915.). The efficacy of such devices to influence clinical symptoms might be different, but the evidence provided by the authors to claim that there is good support for reducing medial tibiofemoral joint contact forces is not 'firm'.
4. Page 4. Line 18. The authors use computational models to estimate the knee force, which is not a 'direct estimation', but rather an indirect one.
5. Page 4 Line 21. It is not clear how the use of the knee adduction moment (KAM) improves the reliability of the results? The KAM provides an 'easy to measure' surrogate of the KMF, but only represents one of 6 inverse dynamic loads and does not account for muscle co-contraction. As such, the KAM is not necessarily a strong predictor of KMF.
6. Page 5. Line 10. It seems like the timing of the increased muscle activation might be important, given these muscles also play a role in developing flexion-extension moments? This might be worth mentioning.

7. Page 6. Line 37. It is unclear how Equation (1) shows a force-endurance relationship from Crowninshield and Brand (1981), as it merely implies a ratio of a maximum muscle force?
8. Page 7. Line 16. What are the 5 vectors of reaction forces? These should be explicitly stated here.
9. Page 8. Line 16. Suggest restructuring sentence.
10. Page 8. Line 22. This is an unusual Euler sequence of rotations. Typically you select the degree of freedom that has the greatest motion as the first rotation (i.e. flexion-extension). Doing otherwise might result in greater kinematic crosstalk.
11. Page 9. Paragraphs 2, 3 and 4. When describing results, the authors commonly state 'significantly larger', however, do not present any statistical evidence for these statements.
12. Page 10. Line 10. It is not clear whether the decreased ankle abduction observed in the experiment resulted directly in a reduction of medial knee contact force, as suggested. This finding represents an association and not a causation.
13. Page 13. Line 3. Using a lower value of c_s in the cost function resulted in an 'obvious reduction in KMF', however, how realistic or physiological would this be?
14. Page 13 line 29. The statement that the ankle adduction angle and KAM 'strengthened the reliability' of the KMF decrease due to latGAS stim is a hypothesis and should maybe be presented in discussion, not the conclusion. Compared to the 'Normal' condition, the change in adduction was only 0.4 degrees (with a SD of 2.1 degrees), which is likely below the accuracy level of the kinematic model.

Author's Response to Decision Letter for (RSOS-171490.R0)

See Appendix A.

RSOS-181545.R0

Review form: Reviewer 1

Is the manuscript scientifically sound in its present form?

No

Are the interpretations and conclusions justified by the results?

No

Is the language acceptable?

Yes

Is it clear how to access all supporting data?

Yes

Do you have any ethical concerns with this paper?

No

Have you any concerns about statistical analyses in this paper?

Yes

Recommendation?

Reject

Comments to the Author(s)

At my request, the authors revised the manuscript quite significantly. They removed two subjects and added some requested information such as GRF.

The muscle forces do affect the joint forces based on the biomechanical structures. However speed, accelerations and ground reaction forces have more significant impact to the knee joint forces. Understandably, high speed would result in higher acceleration, higher ground reaction force and eventually joint force. The biomechanical modelling method might be a good way to quantify the knee joint forces, but a good validation must be presented. The reported work involved some serious questions in the data that concerned me. In the data table 5, the walking speeds in different conditions were recorded as from 0.71 to 0.78, which was 10% variation and quite common due to the difficulties of controlling it. However the first peak of the ground reaction forces from the same four conditions were recorded as identical 0.09, 0.12 and 0.98, which were incomprehensible. Future more, the GRF curves indicated the variations of the four conditions. The vertical GRF from the curves clearly indicated the first peaks were higher than 1.0, no idea why the authors reported as 0.98? Lower than a body mass can hardly be treated as normal walking of normal subjects. I don't know whether to trust table or curves. Before such important questions are clarified, it is hard to discuss the outcome of the paper.

Review form: Reviewer 2**Is the manuscript scientifically sound in its present form?**

Yes

Are the interpretations and conclusions justified by the results?

Yes

Is the language acceptable?

Yes

Is it clear how to access all supporting data?

Yes

Do you have any ethical concerns with this paper?

No

Have you any concerns about statistical analyses in this paper?

No

Recommendation?

Accept with minor revision (please list in comments)

Comments to the Author(s)

The authors have taken on the reviewers comments and made significant changes to the manuscript, which is now easier to understand and the conclusions are within scope of the actual findings. I have only a few minor comments:

1. The description of the normalized force-endurance relationship formula (in response to point 7

of my initial review) was a useful addition for me to understand where this relationship came from. I would suggest including this in the appendix.

2. Page 2 line 50. The authors refer to KAM here as the "knee abduction moment", when I believe it should be the knee ADDUCTION moment (externally applied).

Decision letter (RSOS-181545.R0)

13-Nov-2018

Dear Dr Xu,

The Subject Editor assigned to your paper ("Extra Excitation of Biceps Femoris or Lateral Gastrocnemius during NMES Reduces Knee Medial Loading") has now received comments from reviewers. We would like you to revise your paper in accordance with the referee and Associate Editor suggestions which can be found below (not including confidential reports to the Editor). Please note this decision does not guarantee eventual acceptance.

Please submit a copy of your revised paper before 06-Dec-2018. Please note that the revision deadline will expire at 00.00am on this date. If we do not hear from you within this time then it will be assumed that the paper has been withdrawn. In exceptional circumstances, extensions may be possible if agreed with the Editorial Office in advance. We do not allow multiple rounds of revision so we urge you to make every effort to fully address all of the comments at this stage. If deemed necessary by the Editors, your manuscript will be sent back to one or more of the original reviewers for assessment. If the original reviewers are not available we may invite new reviewers.

When submitting your revised manuscript, you must respond to the comments made by the referees and upload a file "Response to Referees" in "Section 6 - File Upload". Please use this to document how you have responded to each of the comments, and the adjustments you have made. In order to expedite the processing of the revised manuscript, please be as specific as possible in your response.

- Ethics statement

- Data accessibility

It is a condition of publication that all supporting data are made available either as supplementary information or preferably in a suitable permanent repository. The data

accessibility section should state where the article's supporting data can be accessed. This section should also include details, where possible of where to access other relevant research materials such as statistical tools, protocols, software etc can be accessed. If the data has been deposited in an external repository this section should list the database, accession number and link to the DOI for all data from the article that has been made publicly available. Data sets that have been deposited in an external repository and have a DOI should also be appropriately cited in the manuscript and included in the reference list.

If you wish to submit your supporting data or code to Dryad (<http://datadryad.org/>), or modify your current submission to dryad, please use the following link:
<http://datadryad.org/submit?journalID=RSOS&manu=RSOS-181545>

- **Competing interests**

- **Authors' contributions**

- **Acknowledgements**

- **Funding statement**

Please note that Royal Society Open Science charge article processing charges for all new submissions that are accepted for publication. Charges will also apply to papers transferred to Royal Society Open Science from other Royal Society Publishing journals, as well as papers submitted as part of our collaboration with the Royal Society of Chemistry (<http://rsos.royalsocietypublishing.org/chemistry>). If your manuscript is newly submitted and subsequently accepted for publication, you will be asked to pay the article processing charge, unless you request a waiver and this is approved by Royal Society Publishing. You can find out more about the charges at <http://rsos.royalsocietypublishing.org/page/charges>. Should you have any queries, please contact openscience@royalsociety.org.

on behalf of Dr Michael Doube (Associate Editor) and Professor R. Kerry Rowe (Subject Editor)
openscience@royalsociety.org

Associate Editor Comments to Author (Dr Michael Doube):

Dear Dr Xu,

Thank you for your substantial improvements to your manuscript, which have been noted by the reviewers. One reviewer has highlighted a serious discrepancy in the new data that you present, namely Table 5 and Figure 7, and has stated, correctly, that they cannot review the interpretations of the manuscript until the discrepancy is corrected. In particular, the table values of the GRF peaks do not match the peaks in the traces, which casts into doubt the veracity of all the data. Please provide a revision that presents correct and consistent data, and which addresses the reviewer's concerns about validation and that peak GRF < 1 BW cannot represent normal walking.

Best regards,
Michael Doube

Reviewer comments to Author:

Reviewer: 1

Comments to the Author(s)

At my request, the authors revised the manuscript quite significantly. They removed two subjects and added some requested information such as GRF.

The muscle forces do affect the joint forces based on the biomechanical structures. However speed, accelerations and ground reaction forces have more significant impact to the knee joint forces. Understandably, high speed would result in higher acceleration, higher ground reaction force and eventually joint force. The biomechanical modelling method might be a good way to quantify the knee joint forces, but a good validation must be presented. The reported work involved some serious questions in the data that concerned me. In the data table 5, the walking speeds in different conditions were recorded as from 0.71 to 0.78, which was 10% variation and quite common due to the difficulties of controlling it. However the first peak of the ground reaction forces from the same four conditions were recorded as identical 0.09, 0.12 and 0.98, which were incomprehensible. Future more, the GRF curves indicated the variations of the four conditions. The vertical GRF from the curves clearly indicated the first peaks were higher than 1.0, no idea why the authors reported as 0.98? Lower than a body mass can hardly be treated as normal walking of normal subjects. I don't know whether to trust table or curves. Before such important questions are clarified, it is hard to discuss the outcome of the paper.

Reviewer: 2

Comments to the Author(s)

The authors have taken on the reviewers comments and made significant changes to the manuscript, which is now easier to understand and the conclusions are within scope of the actual findings. I have only a few minor comments:

1. The description of the normalized force-endurance relationship formula (in response to point 7 of my initial review) was a useful addition for me to understand where this relationship came from. I would suggest including this in the appendix.

2. Page 2 line 50. The authors refer to KAM here as the "knee abduction moment", when I believe it should be the knee ADDUCTION moment (externally applied).

Author's Response to Decision Letter for (RSOS-181545.R0)

See Appendix B.

RSOS-181545.R1 (Revision)

Review form: Reviewer 1

Is the manuscript scientifically sound in its present form?

No

Are the interpretations and conclusions justified by the results?

No

Is the language acceptable?

Yes

Is it clear how to access all supporting data?

No

Do you have any ethical concerns with this paper?

Yes

Have you any concerns about statistical analyses in this paper?

Yes

Recommendation?

Major revision is needed (please make suggestions in comments)

Comments to the Author(s)

Quite several issues were not properly defined or clarified:

- It seems the gait test had a systematic problem (Table 5). Normally for healthy subjects the first

peak of GRF should be larger than the body weight due to accelerations. When walking is not normal, smaller peak can happen, but the walking would be purposely controlled (limping). Or, to GRF peak 1 less than bodyweight they must be having a very short single support with a long double support which basically means the other limb is taking more of the bodyweight.

I noticed the following explanation of the authors "The peak values mentioned in the manuscript are calculated as the mean values of 20%~35% and 70%~85% of the stance phase, at the same timing as the peak KMFs. Therefore, this mean value is lower than the maximum peak. We have added a clearer explanation in Section 3.4: "The peak GRFs were defined as those at the same timing as the peak KMFs. Specifically, these were calculated as the mean values across 20%-35% and 70%-85% of the stance phase."

However, the mean curve showed clearly the peak 1 is higher than a body weight, then GRFy value could not be less than a body mass as they were listed in the table if the peak1 was calculated as the mean of each peak 1 of each individual trial unless very bad trials were included in the calculation but not in the mean curve. The mean curves of GRF look quite normal and typical. Only the peak 1 needs to be checked.

- The paper didn't report the computation method in detail, such as marker placement and software design, software validation. It is hard for readers to build up their confidence in the results they presented. For example, during normal walking the knee joint force (resultant force) at peak one would not be as high as 1.51 times of body weight and 2.32 times of body weight for medial side of the knee only. I strongly recommend the authors validate their calculation results with other commercial software such as OPEN SIMM or VIUSAL3D (C-Motion, USA) on normal condition and then perform the other calculations.
- Normally KAM is directly proportional to GRFy, a higher GRF peak would produce a higher Kam. The results from this paper showed different results GRFy Peak1=0.97, Peak2=1.08, resulted in the peak 1 and 2 of KAM as 2.07 ± 0.24 1.85 ± 0.60 , which were difficult to comprehend. The model and calculation method should be checked.

- Page 7, line 15: PCSA???

- In Fig 2, In the flow chart of data processing, the inverse dynamic computation was not possible without anthropometer of the subject;

Decision letter (RSOS-181545.R1)

12-Feb-2019

Dear Dr Xu:

On behalf of the Editors, I am pleased to inform you that your Manuscript RSOS-181545.R1 entitled "Extra Excitation of Biceps Femoris during NMES Reduces Knee Medial Loading" has been accepted for publication in Royal Society Open Science subject to minor revision in accordance with the referee suggestions. Please find the referees' comments at the end of this email.

The reviewers and Subject Editor have recommended publication, but also suggest some minor revisions to your manuscript. Therefore, I invite you to respond to the comments and revise your manuscript.

- Ethics statement

If your study uses humans or animals please include details of the ethical approval received, including the name of the committee that granted approval. For human studies please also detail

whether informed consent was obtained. For field studies on animals please include details of all permissions, licences and/or approvals granted to carry out the fieldwork.

- Data accessibility

If you wish to submit your supporting data or code to Dryad (<http://datadryad.org/>), or modify your current submission to dryad, please use the following link:
<http://datadryad.org/submit?journalID=RSOS&manu=RSOS-181545.R1>

- Competing interests

- Authors' contributions

- Acknowledgements

- Funding statement

Because the schedule for publication is very tight, it is a condition of publication that you submit the revised version of your manuscript before 21-Feb-2019. Please note that the revision deadline

will expire at 00.00am on this date. If you do not think you will be able to meet this date please let me know immediately.

on behalf of Dr Michael Doube (Associate Editor) and R. Kerry Rowe (Subject Editor)
openscience@royalsociety.org

Associate Editor Comments to Author (Dr Michael Doube):

Dear Dr Xu,

Thank you for submitting a revised version of your manuscript. One reviewer has some remaining concerns that I believe you will be able to address with minor revisions to your manuscript. Please pay particular attention to clarifying the discrepancy in peak 1 of the GRF curves; to establishing the validity of your computational methods; to the relationship between KAM and GRFy; and to the other points raised by the reviewer. Please supply a point-by-point response for editorial assessment. Please disregard the reviewer's request to compare your software against commercial software; method validation does not require that, but only to ensure that it performs as expected on well-understood data sets. A brief description of how the software was previously validated would help.

Best regards,

Michael Doube

Reviewer comments to Author:

Reviewer: 1

Comments to the Author(s)

Quite several issues were not properly defined or clarified:

- It seems the gait test had a systematic problem (Table 5). Normally for healthy subjects the first peak of GRF should be larger than the body weight due to accelerations. When walking is not normal, smaller peak can happen, but the walking would be purposely controlled (limping). Or, to GRF peak 1 less than bodyweight they must be having a very short single support with a long double support which basically means the other limb is taking more of the bodyweight.

I noticed the following explanation of the authors "The peak values mentioned in the manuscript are calculated as the mean values of 20%~35% and 70%~85% of the stance phase, at the same timing as the peak KMFs. Therefore, this mean value is lower than the maximum peak. We have added a clearer explanation in Section 3.4:"The peak GRFs were defined as those at the same timing as the peak KMFs. Specifically, these were calculated as the mean values across 20%-35% and 70%-85% of the stance phase."

However, the mean curve showed clearly the peak 1 is higher than a body weight, then GRFy value could not be less than a body mass as they were listed in the table if the peak1 was calculated as the mean of each peak 1 of each individual trial unless very bad trials were included in the calculation but not in the mean curve. The mean curves of GRF look quite normal and typical. Only the peak 1 needs to be checked.

- The paper didn't report the computation method in detail, such as marker placement and software design, software validation. It is hard for readers to build up their confidence in the results they presented. For example, during normal walking the knee joint force (resultant force) at peak one would not be as high as 1.51 times of body weight and 2.32 times of body weight for medial side of the knee only. I strongly recommend the authors validate their calculation results with other commercial software such as OPEN SIMM or VIUSAL3D (C-Motion, USA) on normal condition and then perform the other calculations.

- Normally KAM is directly proportional to GRFy, a higher GRF peak would produce a higher Kam. The results from this paper showed different results GRFy Peak1=0.97, Peak2=1.08, resulted

in the peak 1 and 2 of KAM as 2.07 ± 0.24 1.85 ± 0.60 , which were difficult to comprehend. The model and calculation method should be checked.

- Page 7, line 15: PCSA???

- In Fig 2, In the flow chart of data processing, the inverse dynamic computation was not possible without anthropometer of the subject

Author's Response to Decision Letter for (RSOS-181545.R1)

See Appendix C.

Decision letter (RSOS-181545.R2)

18-Feb-2019

Dear Dr Xu,

I am pleased to inform you that your manuscript entitled "Extra Excitation of Biceps Femoris during NMES Reduces Knee Medial Loading" is now accepted for publication in Royal Society Open Science.

on behalf of Dr Michael Doube (Associate Editor) and Professor R. Kerry Rowe (Subject Editor)
openscience@royalsociety.org

Appendix A

Dear Editor and Reviewers,

We are grateful for your detailed review and comments on our manuscript, which have certainly helped to significantly improve our paper. We have replied to all your comments on a point by point basis below.

Yours faithfully,
Authors

Reviewers' Comments to Author:

Reviewer: 1

Comments to the Author(s)

The manuscript is poor in English in the current format. In many cases the present tense and past tense were mixed.

Response:

We apologise for this and have conducted a thorough review of the language used, resulting in a major re-write of the paper.

There are quite some unclarified issues in the models used, for example the meaning of Cs in clinical application, how the KMF acting point was identified? Was it varied like the COP of GRF during stance? Cs was not clarified in the application. Did it have a specific clinical meaning or just a variable used to produce changes in the results.

Response:

We recognise that we have not been clear enough in our description of the model used. In particular, we have now added a clearer description of the meaning of Cs and how the KMF acting point was identified. The following sections of text have been included:

However, as the tibiofemoral joint reaction force is compartmentalized into a medial and a lateral component, the contact points of these two compartments are different from the knee joint center of rotation, and they

were located according to the MR imaging data of the subject mentioned above. The contact points were fixed to the shank segment and defined as the points of action of the medial and lateral knee joint forces; these were the middle of the most distal ends of the femoral condyles in the axial slice. These positions will change on the thigh segment according to the translations between two adjacent segments. The subjects' contact points were obtained by scaling these two points in the MR Imaging data to the subject's shank using the shank scaling factor.

The adjustable weighting factor, c_s , enables the stimulated muscle force to increase relative to standard simulation for a value of c_s of less than 1. Conversely, increasing c_s to greater than 1 will result in under activation of the muscle. As electrical stimulation recruits muscles directly, the combination of voluntary contractions with electrical stimulation produces stronger contraction (40) and so, clinically, c_s is related to stimulation intensity and it is used to produce changes in muscle forces. In order to explore the sensitivity of c_s on the muscle and joint forces, varying values of c_s (1, 0.75, 0.5, 0.25, 0.1 and 0) were implemented. Finally, $c_s=0.1$ for BF and $c_s=0.25$ for latGAS and VL were selected to produce a significant increase in muscle force during the NMES simulation.

The standard deviation of some primary results are too high, the authors didn't present the GRF and the walking speed, which made it hard to assess the quality of the data. Ideally a test and retest reliability should have performed to assure the data quality.

Response:

We have added data on GRF, walking velocity and step length. These are included as Figure 7. Ground reaction force. GRFx: GRF in forward/backward direction; GRFy: GRF in upward/downward direction; GRFz: GRF in rightward/leftward direction, and Table 5. Ground reaction force, walking velocity and step length (mean \pm standard deviation). These are reproduced below.

Figure 7. Components of ground reaction force (GRF) in the forward/backward direction (GRFx); upward/downward direction (GRFy); and right/left direction (GRFz).

Table 5. Ground reaction force, walking velocity and step length (mean \pm standard deviation).

		Normal Walking	loBF Stimulated case	latGAS Stimulated case	VL Stimulated case
GRFx	1st Peak	-0.09 \pm 0.02	-0.09 \pm 0.02	-0.09 \pm 0.02	-0.09 \pm 0.03
(BW)	2nd Peak	0.12 \pm 0.02	0.12 \pm 0.03	0.12 \pm 0.03	0.12 \pm 0.02
GRFy	1st Peak	0.97 \pm 0.02	0.98 \pm 0.04	0.98 \pm 0.03	0.98 \pm 0.04
(BW)	2nd Peak	1.08 \pm 0.06	1.06 \pm 0.06	1.08 \pm 0.05	1.08 \pm 0.06
GRFz	1st Peak	-0.05 \pm 0.01	-0.05 \pm 0.01	-0.05 \pm 0.01	-0.05 \pm 0.01
(BW)	2nd Peak	-0.06 \pm 0.02	-0.06 \pm 0.02	-0.06 \pm 0.02	-0.06 \pm 0.02
	Walking velocity (m/s)	0.78 \pm 0.08	0.71 \pm 0.13	0.75 \pm 0.13	0.77 \pm 0.14
	Step length (m)	0.58 \pm 0.05	0.57 \pm 0.06	0.57 \pm 0.06	0.58 \pm 0.06

Some results are hard to comprehend, such as the knee medial force is directly proportional to the knee adduction moment, however the manuscript presented data curves that showed some contrary results, In the KMF curves the second peak was much higher than the first peak, however in the curves of KAM the first peak was much higher than the second. It is very hard to understand the results as the force should be directly proportional to the knee adduction moment.

Additionally, the KMF and KAM should have the similar variation trend, for example in the KAM curves the KAM of the normal condition was the highest 2nd peak, the others was much lower than it. Contrarily the KMF curves showed the VL Stim was the highest in the second peak, which was hard for readers to understand.

Response:

There is certainly a correlation (shown in the literature) between the knee adduction moment and medial knee force, however, these are not necessarily directly proportional to one another as the equations of motion demonstrate. We have included further comment and discussion on this throughout the manuscript.

Reviewer: 2

Comments to the Author(s)

General Comments

This paper presents an interesting idea, to alter the muscle force distribution

using NMES during walking as a potential intervention for medial tibiofemoral OA. However, there are a number of concerns that limit the impact of this work and the conclusions that can be drawn. These concerns are highlighted below.

The authors assume the use of the knee adduction moment (KAM), as a surrogate measure of the medial force in the knee, as many others also do. However, the KAM is one of 6 inverse dynamic loads and is not necessarily a good predictor of the medial contact force in the knee (KMF), as it does not account for muscle activity across the joint. In the context of this paper, where you are altering muscle forces using NMES would you expect to see a change in the KAM?

Response:

We apologise for the lack of clarity: we have not used KAM as a surrogate measure of the medial force in the knee. We agree that KAM is not a good predictor of KMF, as the relationship between the two parameters is complex. We have included an additional analysis of KAM to help understand the biomechanical change of knee under NMES.

The addition of the parameter, 'c', seems to have the desired effect of changing the muscle force distribution. However, the selection of the parameter value seems a little arbitrary. How is this physiologically related to the relative increase in muscle force from NMES? Can this be estimated by measurement of EMG, or other methods?

Response:

EMG signals are affected by the stimulation artefacts caused by the stimulation current when NMES is used. This has been partially addressed in the literature with, for example, use of an EMG-amplifier with shut-down control, advanced filtering procedures, and optimising the positioning of the EMG electrodes in relation to the stimulation electrodes. However, these approaches do not fully eliminate the stimulation artefacts and we were, therefore, unable to use EMG as a validation of the method. We have added the following:

EMG signals are affected by the stimulation artefacts caused by the stimulation current when NMES is used. This has been partially addressed in the literature with, for example, use of an EMG-amplifier with shut-down control (42), advanced filtering procedures (43), and optimising the positioning of the EMG electrodes in relation to the stimulation electrodes (44). However, these approaches do not fully eliminate the stimulation artefacts and we, therefore, were unable to use EMG as a validation of the

method.

Our selection of the parameter 'c' is therefore arbitrary and we have more carefully articulated this in the manuscript.

Voluntary contractions preferentially recruit type I fibers, and then type II fibers. This asynchronous activation of varied motor units is a physiological mechanism that decreases muscle fatigue (40). However, electrically elicited contractions recruit motor units synchronously, and recruit motor units that are not activated by voluntary contractions (40). It is deduced that stimulated muscles produce larger forces than voluntary conditions, and the other muscles contract in a way to decrease muscle fatigue. In view of this, NMESsim was used with a parameter c to realise a reasonable muscle force distribution: the stimulated muscle force was increased, and all the muscle and joint forces satisfied the motor function constraints. The c_s values were chosen between 0 and 1 (0, 0.1, 0.25, 0.5, 0.75 and 1), where $c_s=1$ represents the original cost function. Where the muscles are not saturated (i.e. limited by their PCSA), assigning a value of c_s of less than 1 enforces a greater muscle force than when using the original cost function; a value of $c_s=0$ assigns the greatest activation to the muscle as this removes the muscle from the cost function.

The changes in the KAM presented in Figure 7 show very large standard deviations and it is interesting that there are statistically significant differences in the 2nd peak of the KAM between the Normal and stim conditions. Although there might be statistical significance, these differences might not be clinically meaningful. The same can be said about the change in ankle adduction angle. A difference of 0.4 deg between the Normal and stim condition does not seem to be clinically meaningful, nor does it appear to be within the accuracy of the measurement system.

Response:

We appreciated the reviewer's comments on clinically meaningful differences and have rewritten our results (we have now included walking velocity and step length in our analysis).

*The mean KAMs and KFMs are shown in Figure 6. The second peak of KAM during loBF stimulated walking is significantly smaller than that during normal walking (1.69 %BW*HT vs. 1.85 %BW*HT, $p=0.016$, Table 4). However, peak KFM does not change with NMES.*

3.4. Ground Reaction Forces, Walking Velocity and Step Length

GRF is presented in Figure 7. There was no statistical difference in peak GRFs, walking velocity and step length between NMES stimulated and normal walking (Table 5).

A comment on the clinical interpretation of this work has been included:

The significant decrease of KAM by loBF stimulation was consistent with that of KMF. A study of 62 OA patients verified the clinical importance of KAM in that peak KAM and KAM impulse explained variance in knee cartilage thickness (47). However, whether the change of KAM could contribute to the alleviation of OA symptoms needs clinical validation.

The authors conclude that NMES provides an opportunity to treat knee osteoarthritis. The changes in KMF due to the stim condition was considered 'not substantial' (page 12, line 57), when compared to some other forms of gait adaptation or gait retraining in the literature. It would be useful to have this discussion in the context of these other modifications. Furthermore, the large variability seen in the KAM results (which reflects the variability you might expect across a patient cohort), demonstrates that perhaps this intervention would work for some individuals, but not others. It would be nice if the simulation could identify which patient might be well-suited to this intervention.

Response:

This is certainly an important point and we have added the following to the discussion:

The decrease in KMF due to NMES is not substantial compared to the reduction caused by other methods. Kinney et al quantified KMF reduction through gait modification using long hiking poles with wide pole placement, and found that KMF at 75% stance decreased significantly from 1.35 BW to 0.89 BW (46). Other modifications in Kinney's study, such as mild crouch and medial thrust, did not show any significant KMF reduction. Orthotic interventions using a lateral foot wedge and valgus knee brace, resulted in a decrease in model-estimated peak tibiofemoral load with increasing wedge size and applied valgus brace moment (7). A higher level of activation using NMES might be required to create a greater reduction in KMF.

Although some of variables were not significantly different between groups, this does not negate the fact that changes can be significant for an

individual. Table 3 lists how peak KMF varies for 8 subjects and although the KMF decrease for the group is not substantial, the simulation could be used to identify which patient is well-suited to this intervention.

There is no discussion regarding the potential changes in the flexion-extension moments following stimulation of each muscle. I also anticipated some discussion regarding the stimulation of combined muscles, which might not change the net joint moment, but together might have a greater impact on KMF than stimulating each muscle independently.

Response:

The following has been added to the discussion:

This study simulated the situations where only one muscle was stimulated. However, it is possible that the stimulation of combined muscles will not change the net joint moment, but together have a greater impact on KMF than stimulating each muscle independently. This case should be considered further.

The muscle forces of loBF, latGAS and VL generate movement of the lower limb mainly through changing the flexion-extension moments of the knee and ankle. The extra muscle activation by NMES may influence the periodic characteristics of normal gait. In this way, different timing of NMES leads to different gait characteristics, and will affect the reduction of KMF. Therefore, NMES timing is an important parameter for precise control of NMES. This study utilised a very crude manually-implemented timing mechanism that should be tuned for optimal outcome.

The authors conclude that the stimulated muscle forces were significantly increased and the KMF was significantly decreased. Without stating the magnitude of these changes, a reader is left with the opinion that these differences were considerable, which is misleading. Furthermore, the evidence to support these statements is not entirely convincing. Firstly, the findings presented regarding muscle force were taken from a model in which the cost function was altered to achieve a desired result of redistributing the muscle force. Secondly, the authors previously state in the discussion that the resulting KMF from the stim conditions were 'not substantial'.

Response:

We have toned down our conclusions and reported the magnitude of the changes::

A new cost function for use in musculoskeletal models, NMESsim, has been proposed to allow the quantification of muscle forces when they are stimulated by external means. This was tested in living subjects, and our results show that the stimulated muscle forces are significantly increased (loBF: $\Delta=0.07$ BW, $p=0.039$; latGAS: $\Delta=0.27$ BW, $p=0.008$; VL: $\Delta=0.40$ BW, $p=0.008$), and the peak values of knee joint medial loading are significantly decreased by applying NMES to loBF (NMESsim: $\Delta=0.17$ BW, $p=0.016$).

Specific Comments

1. Abstract. It would be good to have some quantitative measure of the amount of change in the peak knee medial force, rather than just stating the difference was statistically significant. What was the absolute and/or percentage change? The authors state that there was a strong correlation between knee medial force and knee adduction moment, but again, it would be useful to illustrate the strength of this correlation by supporting the statement with evidence (i.e. an $R^2 = 0.9$).

Response:

Thank you for these suggestions. We have revised the text in the abstract as follows and have made other changes throughout the text:

Stimulation of the biceps femoris resulted in a significant decrease in the second peak of the medial knee joint by up to 0.1 BW ($p=0.016$).

2. Page 3. Line 5. The authors cite a few papers from their own research group here, but this does not really reflect the large efforts that are ongoing amongst the computational biomechanics community to use musculoskeletal modelling to estimate internal forces.

Response:

We have amended this section as follows:

Some musculoskeletal models are driven by electromyography (EMG) measurements, however, NMES stimulus contaminates EMG (27) meaning that this approach cannot be used. Other methods are based on static optimization (28-31) that use cost functions that minimize muscle fatigue (32), defined as the sum of cubed muscles stresses.

27. Lloyd DG, Besier TF. An EMG-driven musculoskeletal model to estimate muscle forces and knee joint moments in vivo. J Biomech. 2003;36(6):765-76.

28. Cleather DJ, Goodwin JE, Bull AM. An optimization approach to inverse

dynamics provides insight as to the function of the biarticular muscles during vertical jumping. Ann Biomed Eng. 2011;39(1):147-60.

29. Li G, Pierce JE, Herndon JH. A global optimization method for prediction of muscle forces of human musculoskeletal system. J Biomech. 2006;39(3):522-9.

30. Heintz S, Gutierrez-Farewik EM. Static optimization of muscle forces during gait in comparison to EMG-to-force processing approach. Gait Posture. 2007;26(2):279-88.

31. Delp SL, Anderson FC, Arnold AS, Loan P, Habib A, John CT, et al. OpenSim: Open-Source Software to Create and Analyze Dynamic Simulations of Movement. IEEE Trans Biomed Eng. 2007;54(11):1940-50.

32. Crowninshield RD, Brand RA. A physiologically based criterion of muscle force prediction in locomotion. J Biomech. 1981;14(11):793-801.

3. Page 3 line 11. In vivo measurement of knee contact forces suggests that there is rather poor evidence of the efficacy of valgus bracing and lateral wedges to reduce knee loading (Kutzner, I., Küther, S., Heinlein, B., Dymke, J., Bender, A., Halder, A. M., & Bergmann, G. (2011). The effect of valgus braces on medial compartment load of the knee joint - in vivo load measurements in three subjects. *Journal of Biomechanics*, 44(7), 1354–1360; and Kutzner, I., Damm, P., Heinlein, B., Dymke, J., Graichen, F., & Bergmann, G. (2011). The effect of laterally wedged shoes on the loading of the medial knee compartment - in vivo measurements with instrumented knee implants. *Journal of Orthopaedic Research*, 29(12), 1910 – 1915.). The efficacy of such devices to influence clinical symptoms might be different, but the evidence provided by the authors to claim that there is good support for reducing medial tibiofemoral joint contact forces is not ‘firm’.

Response:

We have toned down this section to the following:

It has been shown that gait modification by varying the external forces on the knee through valgus bracing and lateral wedges can increase the knee abduction moment (KAM) and/or shift the center of pressure of the ground reaction force laterally (4-7). These measures are surrogates of the medial knee contact forces (KMF) and the efficacy of these approaches is highly dependent on the individual subject (8, 9).

4. Page 4. Line 18. The authors use computational models to estimate the knee force, which is not a ‘direct estimation’, but rather an indirect one.

Response:

The sentence has been rewritten:

Therefore, in this study a modified static optimization cost function is proposed to estimate KMF in NMES-assisted gait.

5. Page 4 Line 21. It is not clear how the use of the knee adduction moment (KAM) improves the reliability of the results? The KAM provides an 'easy to measure' surrogate of the KMF, but only represents one of 6 inverse dynamic loads and does not account for muscle co-contraction. As such, the KAM is not necessarily a strong predictor of KMF.

Response:

We agree with this point and have rewritten this section:

The aim of this study was to test the hypothesis that extra excitation of the lateral leg muscles (i.e. loBF, latGAS and VL) can reduce KMF during level walking, where KMF is quantified through the use of the newly proposed cost function in a validated musculoskeletal model (23). This is then compared with the commonly-used surrogate of KMF, the KAM, and finally, the input kinematic and kinetic data are analyzed to explain the results.

6. Page 5. Line 10. It seems like the timing of the increased muscle activation might be important, given these muscles also play a role in developing flexion-extension moments? This might be worth mentioning.

Response:

We agree that this is certainly potentially important and we have amended the discussion to include the following:

The muscle forces of loBF, latGAS and VL generate movement of the lower limb mainly through changing the flexion-extension moments of the knee and ankle. The extra muscle activation by NMES may influence the periodic characteristics of normal gait. In this way, different timing of NMES leads to different gait characteristics, and will affect the reduction of KMF. Therefore, NMES timing is an important parameter for precise control of NMES. This study utilised a very crude manually-implemented timing mechanism that should be tuned for optimal outcome.

7. Page 6. Line 37. It is unclear how Equation (1) shows a force-endurance relationship from Crowninshield and Brand (1981), as it merely implies a ratio

of a maximum muscle force?

Response:

The normalized force-endurance relationship formula is

$$\log T = -\log\left(\frac{f}{f_{max}}\right)^n + c$$

where T is the endurance time, c is a constant, f is the muscle force, f_{max} is the maximal muscle force and n usually ranges from 2.54 to 3.14. The equation

illustrates that the endurance time increases as the relative muscle force $\left(\frac{f}{f_{max}}\right)^n$ decreases.

The muscle selection to maximize activity endurance is physiologically reasonable during many normal activities. Therefore, the musculoskeletal model to mimic muscle contraction for normal activities has to maximize muscle endurance, which is realized by minimize $\sum\left(\frac{f_i}{f_{imax}}\right)^3$, i.e., the sum of $\left(\frac{f_i}{f_{imax}}\right)^3$ for all muscles.

8. Page 7. Line 16. What are the 5 vectors of reaction forces? These should be explicitly stated here.

Response:

We have added the following:

5 vectors of joint reaction forces for ankle, lateral knee, medial knee, hip and patellofemoral joints.

9. Page 8. Line 16. Suggest restructuring sentence.

Response:

This sentence has been deleted.

10. Page 8. Line 22. This is an unusual Euler sequence of rotations. Typically you select the degree of freedom that has the greatest motion as the first rotation (i.e. flexion-extension). Doing otherwise might result in greater kinematic crosstalk.

Response:

This is a mistake and we apologise for this. We have corrected this in the manuscript:

The sequence of angle calculation is first flexion/extension (plantar flexion/dorsiflexion for the ankle), then external/internal rotation and lastly abduction/adduction.

11. Page 9. Paragraphs 2, 3 and 4. When describing results, the authors commonly state 'significantly larger', however, do not present any statistical evidence for these statements.

Response:

The mean and p values have been added to the text as follows:

There was no significant difference between maximal muscle force values in normal gait and stimulated walking when using the original cost function. However, maximal muscle forces in the stimulated tasks were significantly larger than the maximal muscle forces in normal gait when using NEMSSim for stimulated walking (loBF: 0.27 BW vs. 0.20 BW, $p=0.039$; latGAS: 0.61 BW vs. 0.34 BW, $p=0.008$; VL: 1.18 BW vs. 0.78 BW, $p=0.008$; Table 1).

3.3 Knee Forces and Moments

The mean KMFs are shown in Figure 5. Generally the KMF has two peaks during the stance phase occurring at mean values of 28% and 78% of the stance phase. However, two subjects do not have two explicit peaks. Therefore, the mean KMFs of 20%-35% and 70%-85% of the stance phase are calculated to represent the two peak values. The second peak KMF in loBF stimulated walking is decreased significantly using the original cost function (stimulated vs. normal: 2.26 BW vs. 2.32 BW, $p=0.039$) and NMESSim (stimulated vs. normal: 2.15 BW vs. 2.32 BW, $p=0.016$) in Table 2. The peak KMFs of each subject are shown in Table 3.

*The mean KAMs and KFM are shown in Figure 6. The second peak of KAM during loBF stimulated walking is significantly smaller than that during normal walking (1.69 %BW*HT vs. 1.85 %BW*HT, $p=0.016$, Table 4). However, peak KFM does not change with NMES.*

12. Page 10. Line 10. It is not clear whether the decreased ankle abduction observed in the experiment resulted directly in a reduction of medial knee contact force, as suggested. This finding represents an association and not a causation.

Response:

Due to the very small statistical change, we have decided not to report the joint

angles. (And we agree with the reviewer.)

13. Page 13. Line 3. Using a lower value of c_s in the cost function resulted in an 'obvious reduction in KMF', however, how realistic or physiological would this be?

Response:

This is not known and is a limitation of this manuscript. We have added the following to the discussion:

The decrease of the KMF was limited by the increase of the muscle force, which was determined by the c_s value in the model. Lower c_s values than were applied in this study could be used to generate a greater reduction in KMF. However, the clinical feasibility of including such high muscle forces is not known.

14. Page 13 line 29. The statement that the ankle adduction angle and KAM 'strengthened the reliability' of the KMF decrease due to latGAS stim is a hypothesis and should maybe be presented in discussion, not the conclusion. Compared to the 'Normal' condition, the change in adduction was only 0.4 degrees (with a SD of 2.1 degrees), which is likely below the accuracy level of the kinematic model.

Response:

The analysis of joint angles has been deleted as mentioned above. The conclusion has been revised to the following:

A new cost function for use in musculoskeletal models, NMESsim, has been proposed to allow the quantification of muscle forces when they are stimulated by external means. This was tested in living subjects, and our results show that the stimulated muscle forces are significantly increased (loBF: $\Delta=0.07$ BW, $p=0.039$; latGAS: $\Delta=0.27$ BW, $p=0.008$; VL: $\Delta=0.40$ BW, $p=0.008$), and the peak values of knee joint medial loading are significantly decreased by applying NMES to loBF (NMESsim: $\Delta=0.17$ BW, $p=0.016$). This study demonstrates that it is possible to redistribute the knee loading and reduce the loading on the medial compartment by activating selected muscles across the healthy knee and this opens the door for prevention and alternative non-surgical interventions for knee OA.

Appendix B

Dear Editor and Reviewers,

We are sorry for the unclear description in the manuscript, and grateful for your patience. We reply all your comments on a point by point basis below.

Yours faithfully,

Authors

Reviewer comments to Author:

Reviewer: 1

Comments to the Author(s)

At my request, the authors revised the manuscript quite significantly. They removed two subjects and added some requested information such as GRF.

The muscle forces do affect the joint forces based on the biomechanical structures. However speed, accelerations and ground reaction forces have more significant impact to the knee joint forces. Understandably, high speed would result in higher acceleration, higher ground reaction force and eventually joint force. The biomechanical modelling method might be a good way to quantify the knee joint forces, but a good validation must be presented. The reported work involved some serious questions in the data that concerned me. In the data table 5, the walking speeds in different conditions were recorded as from 0.71 to 0.78, which was 10% variation and quite common due to the difficulties of controlling it. However the first peak of the ground reaction forces from the same four conditions were recorded as identical 0.09, 0.12 and 0.98, which were incomprehensible. Future more, the GRF curves indicated the variations of the four conditions. The vertical GRF from the curves clearly indicated the first peaks were higher than 1.0, no idea why the authors reported as 0.98? Lower than a body mass can hardly be treated as normal walking of normal subjects. I don't know whether to trust table or curves. Before such important questions are clarified, it is hard to discuss the outcome of the paper.

Response:

Dear reviewer, we apologise that some details are missed out.

As you noted at your last review, the KAM showed a very large standard deviation.

We therefore checked the KAM of all the 10 participants (figure 1 below), and found that two of the participants have strange KAM outputs.

Figure 1. KAM of 10 participants. The two dashed lines indicate the abnormal KAM of two participants.

After checking the animation of the kinematic data through the motion analysis software, we noted poor palpation of the pelvic bony landmarks of two participants. These correspond to those with strange data below. This introduced large errors in locating the hip joint centre, and then caused dislocation of femur head (our modelling does not constrain the joints to a fixed centre of rotation), which influenced the result significantly. Therefore, we discarded the data of these two participants.

The peak values mentioned in the manuscript are calculated as the mean values of 20%~35% and 70%~85% of the stance phase, at the same timing as the peak KMFs. Therefore, this mean value is lower than the maximum peak. We have added a clearer explanation in Section 3.4:

“The peak GRFs were defined as those at the same timing as the peak KMFs. Specifically, these were calculated as the mean values across 20%-35% and 70%-85% of the stance phase.”

Reviewer: 2

Comments to the Author(s)

The authors have taken on the reviewers comments and made significant changes to the manuscript, which is now easier to understand and the conclusions are within scope of the actual findings. I have only a few minor comments:

1. The description of the normalized force-endurance relationship formula (in response to point 7 of my initial review) was a useful addition for me to understand

where this relationship came from. I would suggest including this in the appendix.

Response:

Dear reviewer, we have added this part to the Appendix.

“Appendix

The normalized force-endurance relationship formula is

$$\log T = -\log \left(\frac{f}{f_{\max}} \right)^n + c \quad (6)$$

where T is the endurance time, c is a constant, f is the muscle force, f_{\max} is the maximal muscle force and n usually ranges from 2.54 to 3.14. The equation illustrates that the endurance time increases as the relative muscle force $\left(\frac{f}{f_{\max}}\right)^n$ decreases.

The muscle selection to maximize activity endurance is physiologically reasonable during many normal activities. Therefore, the musculoskeletal model to mimic muscle contraction for normal activities has to maximize muscle endurance, which is realized by minimize $\sum \left(\frac{f_i}{f_{imax}}\right)^3$, i.e., the sum of $\left(\frac{f_i}{f_{imax}}\right)^3$ for all muscles.”

2. Page 2 line 50. The authors refer to KAM here as the "knee abduction moment", when I believe it should be the knee ADDUCTION moment (externally applied).

Dear reviewer, we are really sorry to make such a mistake. It should be “knee adduction moment”. We have corrected this in the manuscript.

Appendix C

Dear Editor and Reviewers,

Thank you for your patience. We reply all your comments on a point by point basis below.

Yours faithfully,

Authors

Reviewer comments to Author:

- It seems the gait test had a systematic problem (Table 5). Normally for healthy subjects the first peak of GRF should be larger than the body weight due to accelerations. When walking is not normal, smaller peak can happen, but the walking would be purposely controlled (limping). Or, to GRF peak 1 less than bodyweight they must be having a very short single support with a long double support which basically means the other limb is taking more of the bodyweight.

I noticed the following explanation of the authors “The peak values mentioned in the manuscript are calculated as the mean values of 20%~35% and 70%~85% of the stance phase, at the same timing as the peak KMFs. Therefore, this mean value is lower than the maximum peak. We have added a clearer explanation in Section 3.4: “The peak GRFs were defined as those at the same timing as the peak KMFs. Specifically, these were calculated as the mean values across 20%-35% and 70%-85% of the stance phase.”

However, the mean curve showed clearly the peak 1 is higher than a body weight, then GRFy value could not be less than a body mass as they were listed in the table if the peak1 was calculated as the mean of each peak 1 of each individual trial unless very bad trials were included in the calculation but not in the mean curve. The mean curves of GRF look quite normal and typical. Only the peak 1 needs to be checked.

Response:

Dear reviewer, we apologise that we were not clear enough. In order to make accordance to the calculation of peak KMF, we used the mean values of GRF during 20%~35% and 70%~85% of the stance phase for one trial as the first and second peaks of this trial. For example, here is a GRFy figure of one trial. The stars ‘’ indicate the actual peaks of GRFy. However, we use the mean value of each red part as the peak of GRFy in our paper: the mean of the red part is smaller than the star. Therefore, the peak values listed in Table 5 is certainly smaller than*

the actual peaks, and there are possibilities that the calculated peak 1 is smaller than the body weight.

Reviewer comments to Author:

- The paper didn't report the computation method in detail, such as marker placement and software design, software validation. It is hard for readers to build up their confidence in the results they presented. For example, during normal walking the knee joint force (resultant force) at peak one would not be as high as 1.51 times of body weight and 2.32 times of body weight for medial side of the knee only. I strongly recommend the authors validate their calculation results with other commercial software such as OPEN SIMM or VIUSAL3D (C-Motion, USA) on normal condition and then perform the other calculations.

Response:

Dear reviewer and editor,

We have conducted a lot of work to validate the model, and we have explained the validation of the model in 2.2 Modelling:

“The validation of this original model has been described previously (1-5) through comparison between the model calculations and in-vivo measurements of joint reaction forces or EMG. The revised model with NMES has been validated (5) through the measurement of gluteus maximus activation using EMG pre and post application of NMES on loBF; the kinematics and kinetics input to the model were

measured simultaneously with the EMG, and strong positive correlations were found in early stance peak ($R=0.78$, $p=0.002$) and impulse ($R=0.63$, $p=0.021$).”

Reviewer comments to Author:

- Normally KAM is directly proportional to GRFy, a higher GRF peak would produce a higher Kam. The results from this paper showed different results GRFy Peak1=0.97, Peak2=1.08, resulted in the peak 1 and 2 of KAM as 2.07 ± 0.24 1.85 ± 0.60 , which were difficult to comprehend. The model and calculation method should be checked.

Response:

Dear reviewer, we have checked the model in detail and confirm that the KAM calculated through inverse dynamics can be affected by many factors, such as the GRF, muscle force distribution, the location of knee centre, and so on. It is difficult to estimate KAM by GRFy only and that is why musculoskeletal modelling is required here.

Reviewer comments to Author:

- Page 7, line 15: PCSA???

Response:

We apologise for omitting this. The sentence has been revised to “physiological cross-sectional area (PCSA)”.

Reviewer comments to Author:

- In Fig 2, In the flow chart of data processing, the inverse dynamic computation was not possible without anthropometer of the subject.

Response:

Dear reviewer, we are sorry that the figure was insufficiently clear, so we have redrawn the figure below.

References:

1. Cleather DJ, Bull AM. An optimization-based simultaneous approach to the determination of muscular, ligamentous, and joint contact forces provides insight into musculoligamentous interaction. *Ann Biomed Eng.* 2011;39(7):1925-34.
2. Cleather DJ, Bull AMJ. Lower-extremity musculoskeletal geometry affects the calculation of patellofemoral forces in vertical jumping and weightlifting. *Proc Inst Mech Eng H.* 2010;224(9):1073-83.
3. Cleather DJ, Bull AMJ. The development of a segment-based musculoskeletal model of the lower limb: introducing FREEBODY. *R Soc Open Sci.* 2015;2(6):140449.
4. Ding Z, Nolte D, Kit Tsang C, Cleather DJ, Kedgley AE, Bull AMJ. In Vivo Knee Contact Force Prediction Using Patient-Specific Musculoskeletal Geometry in a Segment-Based Computational Model. *J Biomech Eng.* 2016;138(2):021018--9.
5. Ding Z, Azmi NL, Bull AM. Validation and use of a musculoskeletal gait model to study the role of functional electrical stimulation. *IEEE Trans Biomed Eng.* 2018.